# SELF-CORRECTION VIA TASK DISTILLATION

## ABSTRACT

Large language models (LLMs) have shown promising self-correction abilities, where iterative refinement improves the quality of generated responses. However, most existing approaches operate at the level of output critique, patching surface errors while often failing to correct deeper reasoning flaws. We propose SELF-THOUGHT, a framework that introduces an intermediate step of task abstraction before solution refinement. Given an input and an initial response, the model first distills the task into a structured template that captures key variables, constraints, and problem structure. This abstraction then guides solution instantiation, grounding subsequent responses in a clearer understanding of the task and reducing error propagation. Crucially, we show that these abstractions can be transferred across models: templates generated by larger models can serve as structured guides for smaller LLMs, which typically struggle with intrinsic self-correction. By reusing distilled task structures, smaller models achieve more reliable refinements without heavy fine-tuning or reliance on external verifiers. Experiments across diverse reasoning tasks demonstrate that SELF-THOUGHT improves accuracy, robustness, and generalization for both large and small models, offering a scalable path toward more reliable self-correcting language systems.

## 1 INTRODUCTION

Large Language Models (LLMs) have achieved remarkable progress in reasoning, problem-solving, and dialogue generation (Brown et al., 2020; Chang et al., 2024; Kojima et al., 2022, *inter alia*). However, despite their impressive abilities, even the strongest models often produce errors such as flawed reasoning steps, factual mistakes, or inconsistent results (Maynez et al., 2020; Gehman et al., 2020; Alkaissi & McFarlane, 2023; Yuan et al., 2023, *inter alia*). Self-correction is a capability of LLMs that has recently emerged as a promising solution to mitigate these limitations (Kamoi et al., 2024; Liu et al., 2024, *inter alia*). Recent studies of this and other similar methods (Madaan et al., 2023; Shinn et al., 2023; Welleck et al., 2022; Chen et al., 2024, *inter alia*) show that models can critique their answers, generate feedback, and revise solutions. These methods highlight the promise of intrinsic self-correction, where models improve their own output through iterative refinement.

However, existing self-correction methods (Madaan et al., 2023; Shinn et al., 2023; Cook et al., 2024) have largely take the form of surface-level editing. A model generates an answer, evaluates that answer, and then attempts to patch errors. While effective in some cases, their efficacy in complex problem-solving, such as mathematical reasoning, remains limited. For instance, SELF-REFINE (Madaan et al., 2023) yields an average gain of 20% across tasks, but only modest improvements on mathematical reasoning tasks even when aided by external signals. Similar limitations are observed in SELF-TICK (Cook et al., 2024) and PROGCO (Song et al., 2025), where gains on mathematics and reasoning benchmarks remain marginal compared to the other tasks. Without a structured understanding of the task itself, corrections may be shallow, inconsistent, or fail to generalize beyond the specific example.

Moreover, current self-correction studies are primarily designed for large-scale models, relying on their extensive capacity to generate critiques and perform revisions (Madaan et al., 2023; Cook et al., 2024; Huang et al., 2023; Kamoi et al., 2024). However, these methods often fail to extend to smaller models, which remain widely used in practice due to their efficiency, lower deployment costs, and utility in resource-constrained settings (Kamoi et al., 2024; Madaan et al., 2023; Belcak et al., 2025). Despite their advantages, small models typically lack the reasoning depth and robustness of larger counterparts, and existing self-correction techniques provide little to no measurable improvement

for them. This gap raises an important question of *how to design self-correction mechanisms that are effective not only for frontier LLMs but also for smaller models, enabling them to benefit from iterative refinement*. Addressing this challenge is crucial for broadening the impact of self-correction beyond cutting-edge systems and enabling reliable reasoning across diverse model scales.

In this paper, we propose SELF-THOUGHT, a new framework for iterative self-correction that emphasizes task abstraction before refinement. Instead of immediately critiquing the output, the model first distills the problem into a structured template, identifying variables, constraints, and underlying problem types. This abstraction acts as a reusable guide that grounds subsequent reasoning. The model then instantiates this template to produce a refined solution. By separating understanding the task from solving it, our method reduces error propagation and leads to more robust corrections.

Moreover, we extend SELF-THOUGHT to smaller models through a variant called DISTIL-THOUGHT. In this setting, we reuse the abstract templates distilled by larger, more capable models. These templates encapsulate high-level reasoning and self-correction strategies, allowing smaller models to benefit from structured guidance without requiring external verifiers or costly fine-tuning. By templatizing the problem-solving process, DISTIL-THOUGHT enables smaller models to converge on solutions more quickly and with fewer iterative refinements. This not only improves performance but also offers a cost-saving advantage – reusable templates reduce computational overhead and accelerate inference, making the approach more efficient and scalable across model sizes.

We evaluate our approaches on a range of LLMs, including GPT-4O-MINI, GPT-4O, O3-MINI, DEEPSEEK-R1, and open-source models QWEN-2.5-7B and LLAMA-3.3-70B, across a wide range of tasks. Our findings demonstrate that SELF-THOUGHT consistently surpasses prior techniques, obviating the need for supplementary data or training. For example, when applied to GPT-4O-MINI, SELF-THOUGHT attains $126.30\%$ enhancements on Game of 24, $81.82\%$ gains on Word Sorting, and a $199.85\%$ improvement on AIME 2025. Similarly, on small models such as QWEN-2.5-7B and LLAMA-3.3-70B, DISTIL-THOUGHT yields notable gains, including $154.54\%$ average improvement on QWEN-2.5-7B and $121.42\%$ on LLAMA-3.3-70B, demonstrating that task abstractions learned from large models can effectively transfer to smaller models.

Our primary contributions include: (1) We introduce a two-stage self-correction framework based on task abstraction and solution instantiation, moving beyond surface-level response critique. (2) We demonstrate that abstracted task templates can be reused across models, enabling smaller LLMs to self-correct more effectively. (3) Through experiments across diverse tasks, we show that SELF-THOUGHT improves accuracy, robustness, and generalization compared to baseline self-correction methods. Our work reframes self-correction as a process of thinking about the task rather than fixing the answer. This perspective not only enhances the performance of large models but also provides a scalable path for empowering smaller ones with structured reasoning support.

## 2 RELATED WORK

**Intrinsic Self-Correction.** Intrinsic self-correction seeks to enable models to generate and act on their own feedback during inference, without relying on external signals or additional training. Several methods have been proposed to ask LLMs to critique their initial responses and then attempt refinements (Kim et al., 2023; Shinn et al., 2023; Madaan et al., 2023), but recent studies highlight their limitations, where models often fail to detect reasoning errors reliably, and performance sometimes degrades when self-reflection is applied naively (Huang et al., 2023; Tyen et al., 2024; Kamoi et al., 2024). More structured variants (Shinn et al., 2023; Zelikman et al., 2022) introduce iterative critique or self-distillation, showing that verbal self-feedback can improve output quality, yet these techniques remain brittle and mostly applicable to large models with strong baseline reasoning abilities. In contrast, our method introduces an explicit task abstraction step before refinement, rather than relying on unstructured critique; the model distills the input into a structured template capturing key variables and constraints. This abstraction not only improves the intrinsic correction of large models by grounding refinements in a clearer task representation, but also enables transferability—smaller models, which struggle to generate useful feedback themselves, can leverage abstractions produced by larger models as structured guidance for more reliable correction.

**Source of Feedback.** Feedback is crucial to improve LLM output, with humans traditionally providing corrective signals (Tandon et al., 2021; Elgohary et al., 2021; Bai et al., 2022). Since

---

**Algorithm 1** SELF-THOUGHT algorithm

---

**Require:** Input $x$, model $\mathcal{M}$, prompts $\{\Im, \wp, \Re\}$, stop condition $\mathsf{stop}(\cdot)$, number of iterations $n$
**Ensure:** Corrected output $\hat{y}$ from $\mathcal{M}$
1: Generate initial output $\hat{y}_0 \sim \mathbb{P}_{\mathcal{M}}(\cdot | \Im \oplus x)$ ▷ Initialization
2: **for** iteration $t \in \{0, 1, \ldots, n\}$ **do**
3:     $d_t \sim \mathbb{P}_{\mathcal{M}}(\cdot | \wp \oplus x \oplus \hat{y}_t)$ ▷ Task Abstraction
4:     **if** $\mathsf{stop}(d_t, t)$ **then** ▷ Stopping Criteria
5:        **return** $\hat{y}_t$
6:     **end if**
7:     $\hat{y}_{t+1} \sim \mathbb{P}_{\mathcal{M}}(\cdot | \Re \oplus x \oplus \hat{y}_t \oplus d_t)$ ▷ Instantiation
8: **end for**
9: **return** $\hat{y}_n$

---

human feedback is costly, alternative sources such as scalar reward functions (Bai et al., 2022; Liu et al., 2022; Welleck et al., 2022), external tools like compilers or search engines (Yasunaga & Liang, 2020; Chen et al., 2024; Yu et al., 2023), and domain-specific knowledge bases (Schick et al., 2023) have been used. More recently, LLMs themselves have been employed to generate feedback (Kim et al., 2023; Madaan et al., 2023; Cook et al., 2024), allowing models to iteratively refine their own outputs. However, without structured or verified guidance, LLMs often struggle to correct deeper reasoning errors (Huang et al., 2023). In contrast, our method provides feedback through explicit task abstractions distilled from the input, offering structured guidance. For large models, these abstractions ground refinements in a clearer task representation, while for smaller models they serve as reusable templates from stronger models, enabling more reliable self-correction than unverified self-critiques.

## 3 SELF-THOUGHT: CORRECTING VIA TASK DISTILLATION

Our proposed method, SELF-THOUGHT, builds upon the principle of iterative refinement but introduces a unique intermediate step, *task abstraction*. This abstraction process distills the core components of a problem, transforming it into a structured template that guides the model in generating a more precise and accurate final solution. SELF-THOUGHT first creates a canonical representation of the task itself, ensuring that subsequent solution generation is grounded in a deep understanding of the objectives and constraints of the problem. See Algorithm 1 for a summary of SELF-THOUGHT method, and we describe SELF-THOUGHT in more detail in the following sections.

### 3.1 INITIAL GENERATION.

SELF-THOUGHT generates an initial response $\hat{y}_0$ given an input $x$, prompt $\Im$, and a model $\mathcal{M}$:

$$\hat{y}_0 \sim \mathbb{P}_{\mathcal{M}}(\cdot | \Im \oplus x) \tag{1}$$

where $\Im$ is a task-specific prompt for an initial generation, and $\oplus$ denotes concatenation. This is a standard generation process where the model produces a preliminary response to user input. The initial response serves as the starting point for the subsequent self-correction loop.

### 3.2 TASK ABSTRACTION

Next, SELF-THOUGHT focuses on abstracting the user's input query $x$ and the initial response $\hat{y}_0$ into a structured task representation $d$. This is guided by a structured prompt $\wp$, which facilitates the distillation of the problem into a generalized, constraint-aware template. The model $\mathcal{M}$ is instructed to act as a "meta-distiller" and perform a critical analysis of the user's request. This analysis systematically extracts and formalizes the essential elements of the problem:

$$d_t \sim \mathbb{P}_{\mathcal{M}}(\cdot | \wp \oplus x \oplus \hat{y}_t) \tag{2}$$

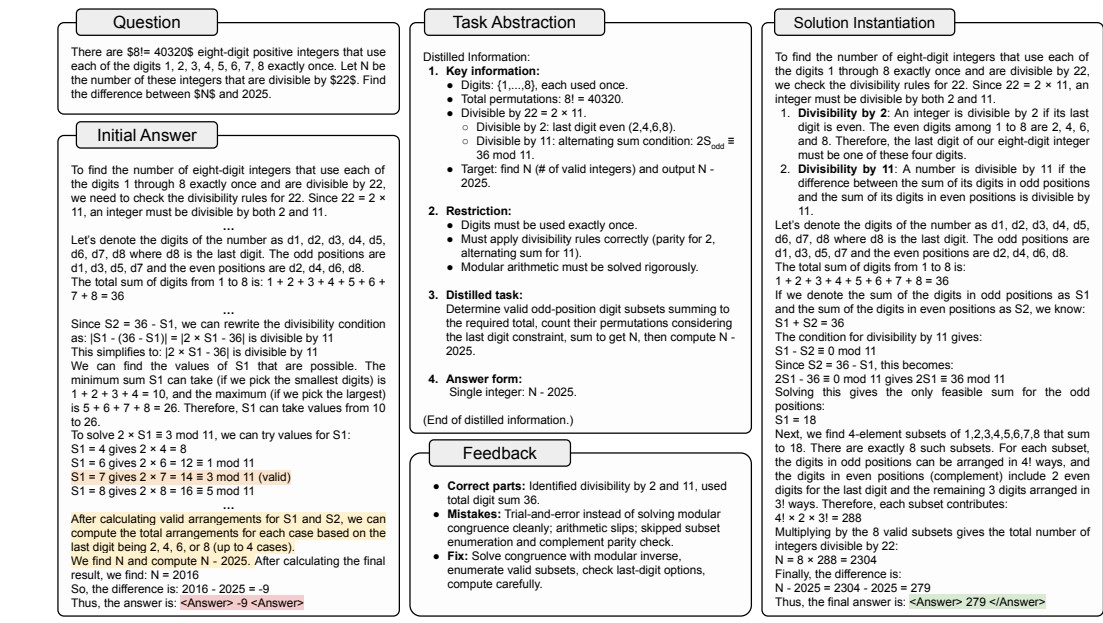

Figure 1: We present an example trace of SELF-THOUGHT self-correcting on a sample from AIME 2025 using GPT-4O-MINI. The initial answer is simplified for clarity, and the full response is provided in Figure 6 in the Appendix. This initial response contains logical reasoning errors, incomplete calculations, and an incorrect final result. By applying task abstraction, SELF-THOUGHT successfully identifies and corrects these mistakes.

The output, $d$, is a structured object that encapsulates the essence of the problem in a format designed to guide subsequent solution generation. An example is presented in Figure 1.

Functionally, as a first step, our method extracts *key information*, identifying all salient variables, values, and data points from $x$. Concurrently, it formalizes the *problem restrictions*, such as mathematical operator precedence or physical laws, to ensure the solution adheres to real-world rules. These explicitly defined constraints are crucial for preventing errors. Finally, SELF-THOUGHT generalizes the problem, reframing $x$ into a higher-level, more abstract, *distilled task* to ensure the solution is robust and applicable to a wider range of similar inputs. Depending on the problem, the structured task abstraction may also contain other constraints such as expected *answer format*.

In concert, these steps encourage the model to identify the underlying problem *type* rather than focusing solely on the specific example. The process also translates the problem into an algorithmic structure, identifying required input parameters and data types, effectively preparing the problem for a programmatic solution. This comprehensive, multi-faceted analysis in the abstraction phase ensures that the final solution in the next phase is grounded in a deep, accurate understanding of the problem's structure and constraints.

### 3.3 SOLUTION INSTANTIATION

Next, SELF-THOUGHT utilizes the distilled information $d$ to generate a specific, concrete solution. The objective here is to instantiate an improved answer ($\hat{y}_{t+1}$) by applying the abstract knowledge of $d$ to the initial query $x$ and response $\hat{y}_t$. The model is provided with the prompt $\Re$ to act as a problem-solving expert to analyze $d$, the input query $x$, and previous output $\hat{y}_t$ to produce a refined and accurate response $\hat{y}_{t+1}$:

$$\hat{y}_{t+1} \sim \mathbb{P}_{\mathcal{M}}(\cdot | \Re \oplus x \oplus \hat{y}_t \oplus d_t) \tag{3}$$

The presence of explicitly defined constraints and the abstracted task in $d$ serves as a powerful guide, significantly reducing the likelihood of errors and ensuring that the solution aligns with the true intent of the problem.

Table 1: A single step of self-correction performance on Game of 24, Word Sorting, Check-mateInOne, AIME 2024, AIME 2025 with GPT-4O-MINI, GPT-4O, O3-MINI, and DEEPSEEK-R1. Green (↑) and red (↓) arrows indicate performance changes against the previous attempt (i.e., INITIAL ($t = 0$)). **Bold** corresponds to the best performance. We find that SELF-THOUGHT consistently yields positive gains between the first and second attempts, demonstrating stable improvements. While baseline approaches often erroneously modify a correct response into an incorrect one, SELF-THOUGHT preserves correctness and consistently improves LLM performance.

| Method | Game of 24 | | | Word Sorting | | | CheckmateInOne | | | AIME 2024 | | | AIME 2025 | | | Mean |
|---|---|---|---|---|---|---|---|---|---|---|---|---|---|---|---|---|
| | Acc@t1 | $\Delta^{i\to c}(t_0,t_1)$ | $\Delta^{c\to i}(t_0,t_1)$ | Acc@t1 | $\Delta^{i\to c}(t_0,t_1)$ | $\Delta^{c\to i}(t_0,t_1)$ | Acc@t1 | $\Delta^{i\to c}(t_0,t_1)$ | $\Delta^{c\to i}(t_0,t_1)$ | Acc@t1 | $\Delta^{i\to c}(t_0,t_1)$ | $\Delta^{c\to i}(t_0,t_1)$ | Acc@t1 | $\Delta^{i\to c}(t_0,t_1)$ | $\Delta^{c\to i}(t_0,t_1)$ | Acc@t1 |
| GPT-4O-MINI | | | | | | | | | | | | | | | | |
| INITIAL ($t=0$) | 38.78 | - | - | 55.0 | - | - | 30.67 | - | - | 20.0 | - | - | 6.67 | - | - | 30.0 |
| REFLEX | 24.49↓14.29 | 8.16 | 22.45 | 60.0↑5.0 | 7.5 | 2.5 | 9.33↓21.34 | 5.33 | 26.67 | 10.0↓10.0 | 0.0 | 10.0 | 10.0↑3.33 | 3.33 | 0.0 | 23.0↓7.0 |
| SELF-REFINE | 25.51↓13.27 | 11.22 | 24.49 | 58.75↑3.75 | 7.5 | 3.75 | 10.67↓20.0 | 6.67 | 26.67 | 13.33↓6.67 | 3.33 | 10.0 | 16.67↑10.0 | 10.0 | 0.0 | 25.0↓5.0 |
| SELF-TICK | 38.78 | 18.37 | 18.37 | 40.0↓15.0 | 7.5 | 22.5 | 20.0↓10.67 | 6.67 | 17.33 | 23.33↑3.33 | 6.67 | 3.33 | 13.33↑6.66 | 6.67 | 0.0 | 27.0↓3.0 |
| REFLEXION | 26.53↓12.25 | 9.18 | 21.43 | 60.0↑5.0 | 11.25 | 6.25 | 9.33↓21.34 | 4.0 | 25.33 | 13.33↓6.67 | 6.67 | 13.33 | 6.67 | 3.33 | 3.33 | 23.0↓7.0 |
| SELF-THOUGHT | **87.76**↑48.98 | 51.02 | 2.04 | **100.0**↑45.0 | 45.0 | 0.0 | **33.33**↑2.66 | 14.67 | 12.0 | **30.0**↑10.0 | 16.67 | 6.67 | **20.0**↑13.33 | 13.33 | 0.0 | **54.0**↑24.0 |
| GPT-4O | | | | | | | | | | | | | | | | |
| INITIAL ($t=0$) | 17.35 | - | - | 86.25 | - | - | 41.33 | - | - | 13.33 | - | - | 10.0 | - | - | 34.0 |
| REFLEX | 19.39↑2.04 | 11.22 | 9.18 | 81.25↓5.0 | 3.75 | 8.75 | 26.67↓14.66 | 12.0 | 26.67 | 13.33 | 0.0 | 0.0 | 6.67↓3.33 | 0.0 | 3.33 | 29.0↓5.0 |
| SELF-REFINE | 33.67↑16.32 | 27.55 | 11.22 | 78.75↓7.5 | 7.5 | 15.0 | 38.67↓2.66 | 13.33 | 16.0 | 20.0↑6.67 | 10.0 | 3.33 | 10.0 | 6.67 | 6.67 | 36.0↑2.0 |
| SELF-TICK | 30.61↑13.26 | 24.49 | 11.22 | 70.0↓16.25 | 3.75 | 20.0 | 30.67↓10.66 | 16.0 | 26.67 | 16.67↑3.34 | 3.33 | 0.0 | 10.0 | 6.67 | 6.67 | 32.0↓2.0 |
| REFLEXION | 36.73↑19.38 | 27.55 | 8.16 | 82.5↓3.75 | 6.25 | 10.0 | 25.33↓16.0 | 16.0 | 32.0 | 16.67↑3.34 | 10.0 | 6.67 | 10.0 | 6.67 | 6.67 | 34.0 |
| SELF-THOUGHT | 37.76↑20.41 | 30.61 | 10.2 | **100.0**↑13.75 | 13.75 | 0.0 | 65.33↑24.0 | 32.0 | 8.0 | 33.33↑20.0 | 20.0 | 0.0 | 16.67↑6.67 | 10.0 | 3.33 | 51.0↑17.0 |
| O3-MINI | | | | | | | | | | | | | | | | |
| INITIAL ($t=0$) | 86.73 | - | - | 90.0 | - | - | 34.67 | - | - | 80.0 | - | - | 73.33 | - | - | 73.0 |
| REFLEX | 83.67↓3.06 | 5.1 | 8.16 | 90.0 | 8.75 | 8.75 | 32.0↓2.67 | 5.33 | 8.0 | 76.67↓3.34 | 3.33 | 3.33 | 76.67↑3.34 | 3.33 | 0.0 | 72.0↓1.0 |
| SELF-REFINE | 86.73 | 11.22 | 11.22 | 87.5↓2.5 | 8.75 | 11.25 | 20.0↓14.67 | 4.0 | 18.67 | 83.33↑3.33 | 10.0 | 6.67 | 73.33 | 6.67 | 6.67 | 70.0↓3.0 |
| SELF-TICK | 0.0↓86.73 | 0.0 | 86.73 | 87.5↓2.5 | 7.5 | 10.0 | 13.33↓21.34 | 2.67 | 24.0 | 76.67↓3.33 | 3.33 | 6.67 | 66.67↓6.66 | 10.0 | 16.67 | 49.0↓24.0 |
| REFLEXION | 84.69↑2.04 | 7.14 | 9.18 | 97.5↑7.5 | 7.5 | 0.0 | 32.0↓2.67 | 12.0 | 14.67 | 80.0 | 3.33 | 3.33 | 66.67↓6.66 | 3.33 | 10.0 | 72.0↓1.0 |
| SELF-THOUGHT | **88.78**↑2.05 | 11.22 | 9.18 | 97.5↑7.5 | 8.75 | 1.25 | **37.33**↑2.66 | 21.33 | 18.67 | **86.67**↑6.67 | 6.67 | 0.0 | **80.0**↑6.67 | 6.67 | 0.0 | **78.0**↑5.0 |
| DEEPSEEK-R1 | | | | | | | | | | | | | | | | |
| INITIAL ($t=0$) | 84.69 | - | - | 97.5 | - | - | 17.33 | - | - | 80.0 | - | - | 63.33 | - | - | 69.0 |
| REFLEX | 64.29↓20.4 | 5.1 | 25.51 | 93.75↓3.75 | 0.0 | 6.25 | 16.0↓1.33 | 9.33 | 10.67 | 76.67↓3.33 | 0.0 | 3.33 | 76.67↓3.33 | 6.67 | 6.67 | 63.0↓6.0 |
| SELF-REFINE | 52.04↓32.65 | 5.1 | 37.76 | 88.75↓8.75 | 1.25 | 10.0 | 16.0↓1.33 | 10.67 | 12.0 | 76.67↓3.33 | 3.33 | 6.67 | 70.0↑6.67 | **20.0** | 13.33 | 61.0↓8.0 |
| SELF-TICK | 17.35↓67.34 | 2.04 | 69.39 | 91.25↓6.25 | 0.0 | 6.25 | 5.33↓12.0 | 1.33 | 13.33 | 60.0↓20.0 | 0.0 | 20.0 | 53.33↓10.0 | 6.67 | 16.67 | 45.0↓24.0 |
| REFLEXION | 50.0↓34.69 | 6.12 | 40.82 | 90.0↓7.5 | 0.0 | 7.5 | 18.67↑1.34 | **14.67** | 13.33 | 56.67↓23.33 | 3.33 | 26.67 | 60.0↓3.33 | 6.67 | 10.0 | 55.0↓14.0 |
| SELF-THOUGHT | 85.71↑1.02 | 12.24 | 11.22 | **100.0**↑2.5 | 2.5 | 0.0 | 20.0↑2.67 | 12.0 | 9.33 | 80.0 | 10.0 | 10.0 | 73.33↑10.0 | 13.33 | 3.33 | 72.0↑3.0 |

The SELF-THOUGHT method thus establishes a self-correction loop where the model's internal analysis of the problem, rather than a critique of its initial output, becomes the mechanism for refinement. This approach ensures that the final response is not merely a corrected version of an initial attempt but a well-reasoned solution derived from a foundational understanding of the problem's structure. This two-step abstraction and instantiation process leads to more robust, reliable, and consistent performance across a wide range of tasks.

### 3.4 TASK DISTILLATION FOR SMALLER MODELS

While SELF-THOUGHT is model-agnostic, we extend its utility to settings where smaller language models struggle with abstraction. In such cases, we leverage the output of the *Task Abstraction* step, $d$, produced by a stronger model. This distilled representation serves as a reusable template that encodes the essential problem structure, constraints, and solution strategy. Given a distilled abstraction $d$ generated by a larger model $\mathcal{M}_L$, a smaller model $\mathcal{M}_S$ can instantiate the solution as:

$$\hat{y}_{t+1}^S \sim \mathbb{P}_{\mathcal{M}_S}(\Re \oplus x \oplus \hat{y}_t^S \oplus d_L \oplus d_S), \tag{4}$$

where $y_t^S$ denotes the current output of the smaller model.

This extension effectively performs *task distillation*: instead of requiring $\mathcal{M}_S$ to perform high-level reasoning from scratch, it inherits the abstract reasoning trace from $\mathcal{M}_L$. As a result, smaller models benefit from the structured guidance in $d$, leading to more accurate self-corrections without incurring the computational overhead of repeatedly prompting larger models.

By decoupling abstraction from instantiation, SELF-THOUGHT not only improves self-correction in a single model but also provides a scalable mechanism for transferring distilled reasoning to less capable models.

## 4 EXPERIMENTAL SETUP

**Datasets.** We evaluate a wide range of tasks that require varying degrees of mathematical and algorithmic reasoning, focusing on problem types where traditional self-correction methods fail. The results and analysis of existing self-correction methods are provided in Appendix G. We conduct

Table 2: A single step of self-correction performance on Game of 24, Word Sorting, CheckmateInOne, AIME 2024, AIME 2025 with *small* models, QWEN-2.5-7B and LLAMA-3.3-70B. Green (↑) and red (↓) arrows indicate performance changes against the previous attempt (i.e., INITIAL ($t = 0$)). **Bold** corresponds to the best performance. Both SELF-THOUGHT and DISTIL-THOUGHT achieve consistent improvements, with the latter leveraging task abstractions.

| Method | Game of 24 | | | Word Sorting | | | CheckmateInOne | | | AIME 2024 | | | AIME 2025 | | | Mean |
|---|---|---|---|---|---|---|---|---|---|---|---|---|---|---|---|---|
| | Acc@t1 | $\Delta^{i\to c}(t_0,t_1)$ | $\Delta^{c\to i}(t_0,t_1)$ | Acc@t1 | $\Delta^{i\to c}(t_0,t_1)$ | $\Delta^{c\to i}(t_0,t_1)$ | Acc@t1 | $\Delta^{i\to c}(t_0,t_1)$ | $\Delta^{c\to i}(t_0,t_1)$ | Acc@t1 | $\Delta^{i\to c}(t_0,t_1)$ | $\Delta^{c\to i}(t_0,t_1)$ | Acc@t1 | $\Delta^{i\to c}(t_0,t_1)$ | $\Delta^{c\to i}(t_0,t_1)$ | Acc@t1 |
| **QWEN-2.5-7B** | | | | | | | | | | | | | | | | |
| INITIAL ($t = 0$) | 8.16 | - | - | 13.75 | - | - | 2.67 | - | - | 20.0 | - | - | 10.0 | - | - | 11.0 - |
| REFLEX | 6.12 ↓2.04 | 5.1 | 7.14 | 16.25 ↑2.5 | 5.0 | 2.5 | 0.0 ↓2.67 | 0.0 | 2.67 | 16.67 ↓3.33 | 3.33 | 6.67 | 10.0 | 0.0 | 0.0 | 10.0 ↓1.0 |
| SELF-REFINE | 6.12 ↓2.04 | 5.1 | 7.14 | 23.75 ↑10.0 | 13.75 | 3.75 | 1.33 ↓1.34 | 1.33 | 2.67 | 20.0 | 10.0 | 10.0 | 3.33 ↓6.67 | 3.33 | 10.0 | 11.0 |
| SELF-TICK | 0.0 ↓8.16 | 0.0 | 8.16 | 17.5 ↑3.75 | 13.75 | 10.0 | 0.0 ↓2.67 | 0.0 | 2.67 | 10.0 ↓10.0 | 3.33 | 13.33 | 6.67 ↓3.33 | 6.67 | 10.0 | 7.0 ↓4.0 |
| REFLEXION | 11.22 ↑3.06 | 9.18 | 6.12 | 21.25 ↑7.5 | 10.0 | 2.5 | 2.67 | 2.67 | 2.67 | 13.33 ↓6.67 | 3.33 | 10.0 | 13.33 ↑3.33 | 6.67 | **3.33** | 12.0 ↑1.0 |
| SELF-THOUGHT | 11.22 ↑3.06 | 8.16 | 5.1 | **66.25** ↑52.5 | 57.5 | 5.0 | 4.0 ↑1.33 | 4.0 | 2.67 | 20.0 | 3.33 | **3.33** | 10.0 | 10.0 | 10.0 | 22.0 ↑11.0 |
| DISTIL-THOUGHT | **41.84** ↑33.68 | 37.76 | 4.08 | 48.75 ↑35.0 | 40.0 | 5.0 | **10.67** ↑8.0 | 10.67 | 2.67 | **23.33** ↑3.33 | 13.33 | 10.0 | **13.33** ↑3.33 | 13.33 | 10.0 | **28.0** ↑17.0 |
| **LLAMA-3.3-70B** | | | | | | | | | | | | | | | | |
| INITIAL ($t = 0$) | 19.39 | - | - | 75.0 | - | - | 8.0 | - | - | 33.33 | - | - | 3.33 | - | - | 28.0 - |
| REFLEX | 42.86 ↑23.47 | 30.61 | 7.14 | 77.5 ↑2.5 | 12.5 | 10.0 | 1.33 ↓6.67 | 0.0 | 6.67 | 36.67 ↑3.34 | 6.67 | 3.33 | 3.33 | 0.0 | 0.0 | 32.0 ↑4.0 |
| SELF-REFINE | 33.67 ↑14.28 | 22.45 | 8.16 | 76.25 ↑1.25 | 11.25 | 10.0 | 5.33 ↓2.67 | 4.0 | 6.67 | 40.0 ↑6.67 | 10.0 | 3.33 | 6.67 ↑3.33 | 3.33 | 0.0 | 32.0 ↑4.0 |
| SELF-TICK | 14.29 ↓5.1 | 12.24 | 17.35 | 71.25 ↓3.75 | 5.0 | 8.75 | 6.67 ↓1.33 | 5.33 | 6.67 | 30.0 ↓3.33 | 6.67 | 10.0 | 6.67 ↑3.33 | 3.33 | 0.0 | 26.0 ↓2.0 |
| REFLEXION | 23.47 ↑4.08 | 19.39 | 15.31 | 76.25 ↑1.25 | 7.5 | 6.25 | 5.33 ↓2.67 | 1.33 | **4.0** | 26.67 ↓6.66 | 3.33 | 10.0 | 3.33 | 0.0 | 0.0 | 27.0 ↓1.0 |
| SELF-THOUGHT | 64.29 ↑44.9 | 48.98 | 4.08 | 98.75 ↑23.75 | 25.0 | 1.25 | 2.67 ↓5.33 | 1.33 | 6.67 | 36.67 ↑3.34 | 10.0 | 6.67 | 16.67 ↑13.33 | 13.33 | 0.0 | 44.0 ↑16.0 |
| DISTIL-THOUGHT | **100.0** ↑80.61 | 80.61 | 0.0 | **100.0** ↑25.0 | 25.0 | 0.0 | **38.67** ↑30.67 | 38.67 | 8.0 | **46.67** ↑13.33 | 20.0 | 6.67 | **23.33** ↑20.0 | 20.0 | 0.0 | **62.0** ↑34.0 |

experiments on **Game of 24** (Yao et al., 2023), **CheckmateInOne** (Srivastava et al., 2023), **Word Sorting** (Suzgun et al., 2023), **AIME 2024** (AIME, 2024), and **AIME 2025** (AIME, 2025). Additional dataset details are provided in Appendix B.1.

**Baselines and Comparison.** We compare our methods to relevant prior approaches based on prompting, sampling, or fine-tuning a single model for both task-solving and self-correction. For prompting, we specifically compare to **REFLEX** (Song et al., 2025), a basic iterative refinement method where the model revises its initial output. **SELF-REFINE** (Madaan et al., 2023) is a representative approach for eliciting self-correction behaviors. **REFLEXION** (Shinn et al., 2023) iteratively evaluates its output, generates verbal feedback, and refines its response based on this feedback. **SELF-TICK** (Cook et al., 2024) generates a checklist of Yes/No questions for the task and uses any unsatisfied points as feedback to improve its output. We include **SELF-CONSISTENCY** (Wang et al., 2023) as a sampling-based baseline, since recent work (Huang et al., 2023) shows it can outperform prompting-based self-correction methods when the number of generated samples matches the number of correction steps. Among the fine-tuning based approaches, we compare to **SUPERCORRECT** (Yang et al., 2025), which enhances small LLM reasoning by distilling thought templates and incorporating self-correction mechanisms, **$S^2R$** (Ma et al., 2025) employs reinforcement learning to teach LLMs to self-verify and self-correct during inference, and **STaSC** (Moskvoretskii et al., 2025) focuses on self-correction for small language models through iterative fine-tuning using solely self-generated data. Complete details of baseline methods are in Appendix B.2.

**Models.** We evaluate our approaches on a diverse set of language models, covering both large and small models. The large models include GPT-4O-MINI (OpenAI, 2024b) and GPT-4O (OpenAI, 2024a), representing strong general-purpose systems.[1] To assess performance in resource-constrained settings, we also consider smaller models such as QWEN-2.5-7B (Qwen, 2025) and LLAMA-3.3-70B (Llama Team, 2024). In addition, we include specialized models focused on reasoning, such as O3-MINI (OpenAI, 2025) and DEEPSEEK-R1 (Guo et al., 2025), which are explicitly designed to handle complex problem-solving tasks.

**Evaluation Protocol and Metrics.** We evaluate performance using different accuracy metrics tailored to the specific requirements of each tasks: **Exact Match (EM)** (Suzgun & Kalai, 2024), which requires the output to match the ground-truth label exactly; **Soft Match (SM)** (Suzgun & Kalai, 2024; Suzgun et al., 2025), which accepts answers containing the correct label while ignoring minor formatting differences; and **Functionally Correct (FC)** (Suzgun & Kalai, 2024; Suzgun et al., 2025), which considers outputs correct if they satisfy task-specific constraints even when formatting or presentation differs. Following prior work (Suzgun & Kalai, 2024; Suzgun et al., 2025; Huang et al., 2023), we use EM for CheckmateInOne, SM for Word Sorting, and FC for Game of 24, AIME 2024, and AIME 2025.

---

[1]We note that newer model families such as GPT-4.1 and GPT-5 were not included, since our experiments were initiated prior to their release, and a full re-evaluation with these models would have incurred considerable additional cost.

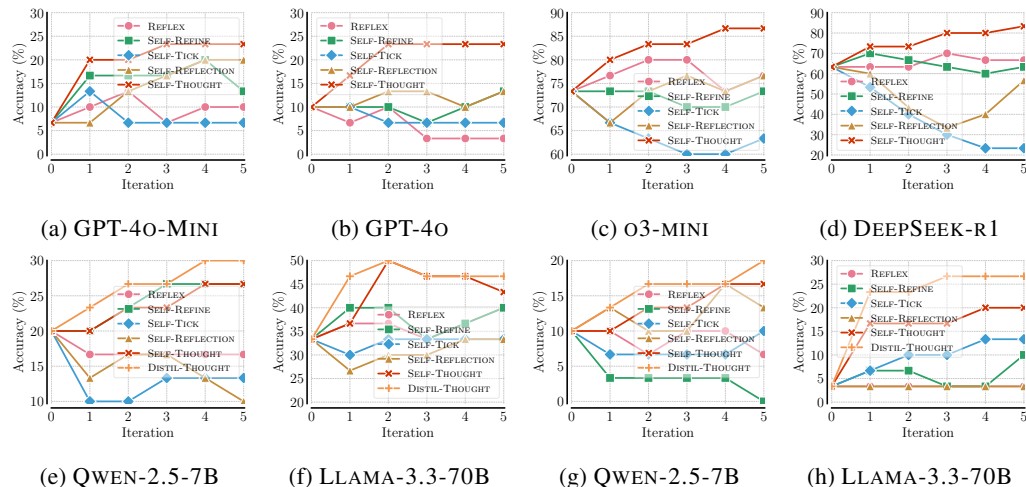

(a) GPT-4O-MINI     (b) GPT-4O     (c) O3-MINI     (d) DEEPSEEK-R1

(e) QWEN-2.5-7B     (f) LLAMA-3.3-70B     (g) QWEN-2.5-7B     (h) LLAMA-3.3-70B

Figure 2: Accuracy over iterations with self-correction methods across models. **Top** row show results on AIME 2024 using large models, while **Bottom** row show results on AIME 2024 (subfigures e and f) and AIME 2025 (subfigures g and h). Please refer to Figures 7 and 8 in the Appendix for the iteration effect plots of other tasks.

To measure self-correction performance, we report and analyze the following metrics: **(1) Acc@ti**: accuracy at the $i$-th attempt; **(2)** $\Delta^{i \to c}(t_{i-1}, t_i)$: the fraction of problems that were incorrect at attempt $i - 1$ but corrected at attempt $i$, capturing how many new problems self-correction solves; and **(3)** $\Delta^{c \to i}(t_{i-1}, t_i)$: the fraction of problems that were correct at attempt $i - 1$ but become incorrect at attempt $i$, reflecting how reliably the model preserves correct answers.

**Selecting Task Abstraction.** For our experiments, we randomly sample a single task abstraction from the set of successful cases, i.e., those in which the large model (GPT-4O-MINI) successfully corrected the initial output. Our idea is that abstractions associated with successful corrections provide at least one concrete example of reasoning that leads to the right solution, thereby offering a useful, though not necessarily optimal, guideline for subsequent models. The selected abstraction is then reused across smaller models, serving as structured guidance to support their self-correction. This choice allows us to test whether relatively lightweight abstractions, distilled from a mid-sized model, are sufficient to enhance the performance of smaller models without relying exclusively on the largest and most costly systems.

## 5 EXPERIMENTS AND RESULTS

**Main Results.** Table 1 shows the self-correction results across five reasoning benchmarks and four different models. We observe that SELF-THOUGHT consistently yields the highest accuracy after one round of self-correction, outperforming other intrinsic methods such as SELF-REFINE, SELF-TICK, and REFLEXION. We show results on iterative self-correction in the experiments, with additional analysis in Appendix F.3. For instance, on GPT-4O-MINI, SELF-THOUGHT improves **Acc@t1** from 38.78% to 87.76% on Game of 24 and from 55.0% to 100.0% on Word Sorting, corresponding to gains of ↑48.98% and ↑45.0%, respectively. In comparison, SELF-REFINE and REFLEXION show far smaller net improvements and often increase $\Delta^{c \to i}(t_0, t_1)$, indicating that they mistakenly alter correct responses. This trend is most evident in reasoning-heavy tasks such as CheckmateInOne, where SELF-THOUGHT consistently improves performance across all models, while competing approaches often reduce accuracy by altering correct answers into incorrect ones.

Looking across models, general models such as GPT-4O-MINI and GPT-4O benefit the most from SELF-THOUGHT. On GPT-4O, it raises mean **Acc@t1** from 34.0% (INITIAL ($t = 0$)) to 51.0%, with especially large gains on CheckmateInOne and AIME 2024. Reasoning models like O3-MINI and DEEPSEEK-R1, which already start from stronger baselines, still see consistent positive gains, SELF-THOUGHT improves mean **Acc@t1** from 73.0% to 78.0% on O3-MINI and from 69.0% to

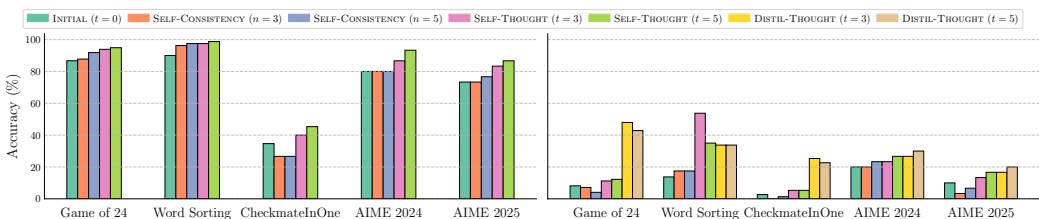

Figure 3: Comparison of SELF-THOUGHT and DISTIL-THOUGHT with the SELF-CONSISTENCY on O3-MINI and QWEN-2.5-7B. See Figure 5 in Appendix E for the results on other models.

72.0% on DEEPSEEK-R1. Importantly, the balance between $\Delta^{i\to c}(t_0, t_1)$ and $\Delta^{c\to i}(t_0, t_1)$ confirms that SELF-THOUGHT encourages conservative but effective revisions. For example, on GPT-4O, it achieves $\Delta^{i\to c}(t_0, t_1)$ of 13.75% on Word Sorting while keeping $\Delta^{c\to i}(t_0, t_1)$ zero, unlike SELF-TICK, which achieves a high $\Delta^{i\to c}(t_0, t_1)$ but with much higher $\Delta^{c\to i}(t_0, t_1)$. Taken together, these results show that SELF-THOUGHT reliably improves self-correction across tasks and scales, whereas prior intrinsic methods either provide limited benefits or destabilize performance by introducing unnecessary changes.

**Results on Small Models.** Table 2 shows the results on QWEN-2.5-7B and LLAMA-3.3-70B. We observe that baseline self-correction methods often fail to deliver consistent gains and, in most cases, even degrade performance. For instance, on QWEN-2.5-7B, REFLEX reduces mean accuracy from 20.0 at initialization to 10.0, while SELF-TICK drops it further to 7.0. A similar pattern holds for LLAMA-3.3-70B, where SELF-TICK lowers mean accuracy from 28.0 to 26.0, and SELF-REFINE shows only modest improvements to 32.0. These trends highlight that small models struggle to generate useful intrinsic feedback, often flipping correct answers into incorrect ones.

In contrast, both SELF-THOUGHT and especially DISTIL-THOUGHT achieve substantial improvements by leveraging task abstractions distilled from larger models. On QWEN-2.5-7B, DISTIL-THOUGHT raises mean accuracy from 11.0 to 28.0, outperforming all baselines, while SELF-THOUGHT provides a moderate improvement to 22.0. The effect is even more striking for LLAMA-3.3-70B, where DISTIL-THOUGHT boosts mean accuracy from 28.3 to 62.0, more than doubling performance, with particularly large gains on reasoning-heavy tasks such as AIME 2024 (from 33.0 to 46.67) and AIME 2025 (from 3.33 to 23.33). These results demonstrate that abstraction transfer offers small models a reliable pathway to self-correction, bridging the gap between their limited reasoning ability and the stronger feedback signals required for improvement.

**Effect of Iterative Correction.** We examine the effect of iterative correction for all tasks using different models. The results on AIME 2024 and AIME 2025 are depicted in Figure 4, with more results on other tasks provided in Appendix F.3. We find that iterative self-correction consistently improves accuracy across models, with the largest gains in early iterations. Importantly, this early-stage improvement also translates to reduced computational cost, as models require fewer refinement cycles to reach high-quality solutions. DISTIL-THOUGHT achieves the highest correct-flip proportions, outperforming baseline self-correction methods such as SUPERCORRECT and S²R by a notable margin ($10 - 15\%$ improvement in early rounds). While later iterations continue to provide improvements, the marginal gains diminish after 2–3 rounds. Additionally, model-only self-correction without external guidance or feedback is less effective (e.g., using external tools for feedback (Gou et al., 2024)), showing slower convergence and lower overall gains compared to approaches leveraging structured iterative updates.

**Comparison with SELF-CONSISTENCY.** Recent work by Huang et al. (2023) indicates that SELF-CONSISTENCY outperforms many existing self-correction strategies, such as multi-agent debate (Du et al., 2023; Liang et al., 2024; Chen et al., 2023), when applied under the same number of response samples. We therefore adopt SELF-CONSISTENCY (Wang et al., 2023) as an additional baseline for comparison with our proposed methods. SELF-CONSISTENCY generates multiple candidate responses and selects the final output through majority voting. We evaluate this method using $n \in \{3, 5\}$ samples, aligning with the number of self-correction iteration used in

Table 3: Results of correct and incorrect flips, comparing with fine-tuning based models on AIME 2024 and AIME 2025. See Table 4 in the Appendix for detailed results on all evaluation tasks.

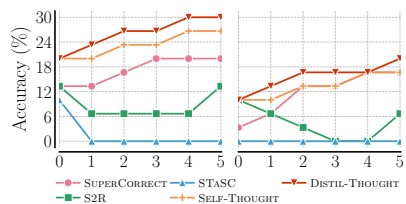

Figure 4: Accuracy over iterations on (**Right**) AIME 2024 and (**Left**) AIME 2025.

| Method | Iteration 1 | | Iteration 2 | | Iteration 3 | | Iteration 4 | | Iteration 5 | |
|---|---|---|---|---|---|---|---|---|---|---|
| | $\Delta^{i \to c}(t_0,t_1)$ | $\Delta^{c \to i}(t_0,t_1)$ | $\Delta^{i \to c}(t_1,t_2)$ | $\Delta^{c \to i}(t_1,t_2)$ | $\Delta^{i \to c}(t_2,t_3)$ | $\Delta^{c \to i}(t_2,t_3)$ | $\Delta^{i \to c}(t_3,t_4)$ | $\Delta^{c \to i}(t_3,t_4)$ | $\Delta^{i \to c}(t_4,t_5)$ | $\Delta^{c \to i}(t_4,t_5)$ |
| **AIME 2024** | | | | | | | | | | |
| SUPERCORRECT | 0.0 | **0.0** | 3.33 | 0.00 | 3.33 | 0.00 | 0.00 | 0.00 | 0.00 | 0.00 |
| S²R | 3.33 | 10.00 | 0.00 | 0.00 | 3.33 | 3.33 | 3.33 | 3.33 | **10.00** | 3.33 |
| STASC | 0.00 | 10.00 | 0.00 | 0.00 | 0.00 | 0.00 | 0.00 | 0.00 | 0.00 | 0.00 |
| SELF-THOUGHT | 3.33 | 3.33 | 3.33 | 0.00 | 0.00 | 0.00 | 0.00 | 3.33 | 0.00 | 0.00 |
| DISTIL-THOUGHT | **13.33** | 10.00 | 3.33 | 0.00 | 3.33 | 3.33 | **6.67** | 3.33 | 0.00 | 0.00 |
| **AIME 2025** | | | | | | | | | | |
| SUPERCORRECT | 3.33 | 0.00 | 6.67 | 0.00 | 0.00 | 0.00 | 3.33 | 0.00 | 3.33 | 3.33 |
| S²R | 3.33 | 6.67 | 3.33 | 6.67 | 0.00 | 3.33 | 0.00 | 0.00 | **6.67** | 0.00 |
| STASC | 0.00 | 0.00 | 0.00 | 0.00 | 0.00 | 0.00 | 0.00 | 0.00 | 0.00 | 0.00 |
| SELF-THOUGHT | 10.00 | 10.00 | 3.33 | 0.00 | 3.33 | 3.33 | 3.33 | 0.00 | 0.00 | 0.00 |
| DISTIL-THOUGHT | **13.33** | 10.00 | 3.33 | 0.00 | **6.67** | 6.67 | 0.00 | 0.00 | 3.33 | 0.00 |

our methods. Figure 3 shows the results across O3-MINI and QWEN-2.5-7B on five reasoning tasks. On O3-MINI, SELF-CONSISTENCY shows moderate gains over the initial responses, especially on Word Sorting, AIME 2024, and AIME 2025. However, our methods consistently match or surpass SELF-CONSISTENCY. For example, on CheckmateInOne, where SELF-CONSISTENCY only marginally improves performance, SELF-THOUGHT achieves a notable increase in accuracy, indicating its ability to better exploit intermediate reasoning. Similarly, on AIME 2024 and AIME 2025, SELF-THOUGHT with $t = 5$ outperforms SELF-CONSISTENCY, highlighting the effectiveness of structured task distillation in improving final answers. The difference is more noticeable on QWEN-7B. On small models, SELF-CONSISTENCY provides limited improvement, and in some cases (e.g., CheckmateInOne), it remains close to the baseline. In contrast, both SELF-THOUGHT and DISTIL-THOUGHT yield substantial gains. For instance, DISTIL-THOUGHT with $t = 5$ boosts accuracy by more than 20 points on Game of 24 and by over 15 points on Word Sorting compared to SELF-CONSISTENCY, showing the scalability of our methods even for small models. These results suggest that while SELF-CONSISTENCY can provide benefits through sampling, our approaches more effectively harness reasoning traces, yielding stronger and more stable improvements across diverse tasks.

**Comparison with Fine-Tuning Baselines.** Table 3 and Figure 4 show the comparison results of our methods with fine-tuned based self-correction methods on AIME 2024 and AIME 2024, with more results provided in Appendix D. Our methods, SELF-THOUGHT and DISTIL-THOUGHT, achieve clear gains over all fine-tuning based baselines. DISTIL-THOUGHT shows the strongest trends, with steady accuracy growth and the highest correct-flip rates (e.g., $13.3\%$ in iteration 1), while SELF-THOUGHT yields consistent improvements up to $16\%$ accuracy on AIME 2025. In contrast, S²R and STASC plateau or decline, and SUPERCORRECT stays flat. Notably, many baselines are fine-tuned on math datasets for self-correction and benefit from training on task, whereas our methods obtain strong generalization without heavy in-domain supervision. Overall, this highlights the efficiency of our lightweight self-correction strategies compared to costly fine-tuning approaches.

# 6 CONCLUSION

We introduce SELF-THOUGHT and DISTIL-THOUGHT, two complementary approaches for enhancing self-correction in language models. SELF-THOUGHT empowers models to refine their reasoning through structured introspection, while DISTIL-THOUGHT leverages distilled guidance from a larger model to improve correction efficiency and stability. Together, they offer a lightweight yet effective alternative to existing prompting- and fine-tuning-based methods. Our experiments across diverse reasoning benchmarks demonstrate consistent improvements in correction accuracy and flip reliability, especially in early iterations, highlighting both the generality and robustness of our framework. Beyond performance, our analysis sheds light on the limitations of current self-correction techniques and underscores the value of structured reasoning templates for building more trustworthy systems.

## REPRODUCIBILITY STATEMENT

We run all experiments on public benchmarks. We use publicly accessible large language models for evaluation, API access to GPT models is available at `https://openai.com/api`, and we use HuggingFace for open-source models.

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

CONTENTS

## A    LIMITATIONS & FUTURE WORK

**Task Abstraction Selection.**    Our current approach relies on randomly selecting task abstractions from successful GPT-4O-MINI cases. While this provides a practical starting point, it does not guarantee that the chosen abstraction is the most effective one for a given problem. Future work could explore more principled strategies, such as embedding-based similarity measures to align problems with the most relevant abstractions. Additionally, abstractions could be generated from stronger models (e.g., GPT-4O, O3-MINI, or DEEPSEEK-R1), which may yield richer reasoning patterns. A systematic comparison across source models would clarify whether larger and more capable models produce abstractions that better generalize across tasks.

## B    ADDITIONAL EXPERIMENT DETAILS

### B.1    DATASETS AND TASKS

To evaluate the efficacy of our proposed approach compared to other state-of-the-art proposed self-correction baselines, we consider a wide range of tasks and datasets that require various degrees of mathematical and algorithmic reasoning. The introduction to the evaluation datasets is as follows:

- **Game of 24** (Yao et al., 2023): A mathematical reasoning challenge where the objective is to form an expression that evaluates to $24$ using four given numbers exactly once. For instance, if the input values were "7 7 8 11," one valid answer would be "$8 * (7 + 7 - 11)$." This task emphasizes systematic search, strategic reasoning, and pattern recognition. We use the 99 examples from (Yang et al., 2024) to evaluate models capacity for refining computational heuristics and strategy over manual attempts.

- **CheckmateInOne** (Srivastava et al., 2023): A chess reasoning challenge where the objective is to identify the move, expressed in Standard Algebraic Notation (SAN), that delivers checkmate in a given position. The input consists of a sequence of prior moves leading to a state where a one-move checkmate is possible. For instance, after the sequence "1. e4 e5 2. Qh5 Nc6 3. Bc4 Nf6," the correct output is "Qxf7#." This task probes spatial reasoning, rule application, and tactical foresight. We use 3,500 curated game positions to evaluate models' ability to achieve exact match accuracy in identifying checkmating moves.

- **Word Sorting** (Suzgun et al., 2023): A linguistic reasoning challenge where the model must sort a given list of words according to a specified criterion, such as alphabetical order, length, or semantic attributes. For example, sorting "cat, elephant, dog" by length yields "cat, dog, elephant." This task tests systematic application of sorting rules, attention to fine-grained instructions, and consistency in following multi-step language-based procedures.

- **AIME 2024 and AIME 2025**: The American Invitational Mathematics Examination (AIME) is a prestigious high-school competition featuring complex problems across algebra, combinatorics, number theory, geometry, and probability. These questions require deep mathematical reasoning and multi-step problem-solving. We consider two subsets that are shown to be challenging for large language models (Suzgun et al., 2025), namely, AIME 2024[2] and AIME 2025[3], where each subset has 30 questions.

### B.2    BASELINES

Here, we introduce the details of the baseline methods for comparison with our proposed method:

- **REFLEX**: REFLEX is a basic iterative refinement method in which the LLM reflects on its initial output and generates a revised response. We include REFLEX as one of our comparison baselines, following the recent work of Song et al. (2025), which showed that this simple approach can improve performance when using large base models such as GPT-4o (refer to Table 1 in Song et al. (2025) for detailed results).

---

[2] https://huggingface.co/datasets/HuggingFaceH4/aime_2024
[3] https://huggingface.co/datasets/yentinglin/aime_2025

- **SELF-REFINE** (Madaan et al., 2023): SELF-REFINE iteratively reviews its own output to generate feedback and proposes refinements based on the feedback from the previous step, continuing this process until no errors are detected or a maximum number of iterations is reached.

- **REFLEXION** (Shinn et al., 2023): REFLEXION is an iterative approach where the model first evaluates its output, then generates verbal feedback about its previous output based on the evaluation and uses this feedback to refine its output. Shinn et al. (2023) uses ground truth labels about answer correctness for evaluation to guide the self-correction process. However, in our experiments, we rely on the Chain-of-Thought (CoT) generated by the model itself, because we assume that the ground truth context or an external API is not available (see Section 4.2 in Shinn et al. (2023)).

- **SELF-TICK** (Cook et al., 2024): SELF-TICK first generates a checklist, i.e., Yes/No questions, for the input task, and then verifies whether the generated response satisfies all the questions from the checklist one by one. Any unsatisfied verification points will be used as feedback to refine and improve its own output.

- **SELF-CONSISTENCY** (Wang et al., 2023): SELF-CONSISTENCY is a decoding strategy that samples diverse reasoning trajectories from the model and selects the most consistent answer based on majority voting among these reasoning traces.

- **SuperCorrect** (Yang et al., 2025): SuperCorrect is a two-stage framework in which a large teacher model supervises the reasoning and self-correction processes of a smaller student. First, the reasoning trajectories generated by the teacher model are used to perform Supervised Fine-Tuning (SFT) on the student model to enhance its reasoning capabilities. Then, the teacher model provides corrections for the hierarchical reasoning trajectories generated by the SFT-fine-tuned student model, and a collaborative Direct Preference Optimization (DPO) technique is applied to improve the ability of the student model to refine its outputs based on these correction traces. Yang et al. (2025) used `o1-mini` or `GPT-4o-mini` as the teacher model and `Qwen-2.5-Math-7B-Instruct` as the student model.

- **$S^2R$** (Ma et al., 2025): $S^2R$ introduces a framework that enhances LLM reasoning by teaching models to self-verify and self-correct during inference. It begins by initializing LLMs with iterative self-verification and self-correction behaviors through supervised fine-tuning on curated data. These skills are further strengthened by both outcome-level and process-level reinforcement learning, with minimized resource requirements, enabling the model to adaptively refine its reasoning process during inference. Experimental results demonstrate significant accuracy improvements, showcasing the effectiveness of $S^2R$ in enhancing LLM reasoning capabilities.

- **STaSC** (Moskvoretskii et al., 2025): STaSC focuses on self-correction in small language models through iterative fine-tuning using solely self-generated data. The Self-Taught Self-Correction (STaSC) algorithm incorporates multiple algorithmic design choices, allowing models to improve their outputs without external supervision. Experimental results on a question-answering task demonstrate that STaSC effectively learns self-correction, leading to significant performance improvements. The study provides insights into the mechanisms of self-correction and the impact of different design choices on learning dynamics and overall performance.

### B.2.1 COMPRESSION AGAINST SELF-CONSISTENCY

Why should we compare our approach with SELF-CONSISTENCY (Wang et al., 2023)? SELF-CONSISTENCY prompts models to generate multiple responses and select the most consistent responses by performing majority voting. A recent study (Huang et al., 2023) shows that SELF-CONSISTENCY outperforms the multi-agent debate approach with the equivalent number of responses. A recent study compared majority voting with the other techniques and showed that it outperforms other aggregation functions (Song et al., 2025).

### B.2.2 COMPRESSION AGAINST SUPERCORRECT

Why should we compare our approach with SUPERCORRECT (Yang et al., 2025)? SUPERCORRECT is a recent self-correcting model that leverages distillation from larger models. A model that is fine-

tuned in two stages, SFT and DPO. In the SFT stage, the model is fine-tuned on reasoning traces generated by a larger model on math datasets, and in the DPO stage, it is fine-tuned on a preference pair dataset of corrected reasoning trajectories generated by the large model and the small model.

### B.3 ANSWER EXTRACTION PROTOCOL

To keep the evaluation consistent and reliable, all models are asked to write their final answers in a structured and machine-readable format. Each answer is expected to be wrapped in the following XML-style tags:

<**Answer**> Your Final Answer Here </**Answer**>

This specific format makes it easy to correctly read and process the answers, avoiding mistakes from extra text or ambiguous outputs. After being extracted, the final answers are evaluated using the accuracy measure for each specific task.

### B.4 EVALUATION PROTOCOL

Given the diversity of tasks, we use different accuracy metrics tailored to the specific requirements of each benchmark:

- **Exact Match (EM)**. EM is a strict metric that marks an answer as correct only if it matches the ground-truth label exactly, without extra text or formatting differences.
- **Soft Match (SM)**. SM is a lenient metric that marks an answer as correct if the ground-truth label appears in the model's output, ignoring minor formatting differences such as punctuation or whitespace. Unlike EM, SM does not require the output to match the label verbatim.
- **Functionally Correct (FC)**. FC is a flexible metric that marks an answer as correct if it satisfies task-specific constraints, even when the exact wording, numeral presentation, or formatting differs from the reference solution.

We apply **EM** for CheckmateInOne, **SM** for Word Sorting, and **FC** for Game of 24, AIME 2024, and AIME 2025 benchmarks.

To measure self-correction performance, we report and analyze the following metrics: **(1) Acc@ti**: accuracy at the $i$-th attempt; **(2)** $\Delta^{i\rightarrow c}(t_{i-1}, t_i)$: the fraction of problems that were incorrect at attempt $i-1$ but corrected at attempt $i$, capturing how many new problems self-correction solves; and **(3)** $\Delta^{c\rightarrow i}(t_{i-1}, t_i)$: the fraction of problems that were correct at attempt $i-1$ but become incorrect at attempt $i$, reflecting how reliably the model preserves correct answers.

## C PROMPTS

### C.1 INITIAL GENERATION

---

**Game of 24**

Let's play a game called 24. You'll be given four integers, and your objective is to use each number only once, combined with any of the four arithmetic operations (addition, subtraction, multiplication, and division) and parentheses, to achieve a total of 24. For example, if the input is 4, 7, 8, and 8, the output could be (7 * 8) - (4 * 8). You only need to find one feasible solution!
Input: {*question*}. Please provide the final answer within <Answer> Your Final Answer Here </Answer>.

---

**Word Sorting**

Sort a list of words alphabetically, placing them in a single line of text separated by spaces. Input: {*question*}. Please provide the final answer within <Answer> Your Final Answer Here </Answer>.

**CheckmateInOne**

Given a series of chess moves written in Standard Algebraic Notation (SAN), determine the next move that will result in a checkmate.
Input: {*question*}. Please provide the final answer within <Answer> Your Final Answer Here </Answer>.

**AIME 2024**

Given the input question, your task is to provide the answer to the question.
Input: {*question*}. Please provide the final answer within <Answer> Your Final Answer Here </Answer>.

**AIME 2025**

Given the input question, your task is to provide the answer to the question.
Input: {*question*}. Please provide the final answer within <Answer> Your Final Answer Here </Answer>.

## C.2 TASK ABSTRACTION

---

### Task Abstraction and Distillation

As a highly professional and intelligent expert in information distillation, you excel at extracting essential information to solve problems from user input queries. You adeptly transform this extracted information into a suitable format based on the respective type of issue. If the problem can be generalized to a higher level to solve multiple issues, further analysis and explanation will be provided upon your next response. Please categorize and extract the crucial information required to solve the problem from the user's input query. Combining these two elements will generate distilled information. The distilled information should include:

1. Values and information of key variables extracted from user input, which will be handed over to the respective expert for task resolution, ensuring all essential information required to solve the problem is provided.
2. The objective of the problem and corresponding constraints.
3. Extend the problem based on 1 and 2, propose a meta problem that can address the user query and handle more input and output variations. Incorporate the real-world scenario of the extended problem along with the types of key variables and information constraints from the original problem to restrict the key variables in the extended problem. After that, use the user query input key information as input to solve the problem, as an example.
4. Try to transform the problem into a Python algorithm problem, and provide the input parameters.
5. Your task is to distill the problem; you shouldn't give the final result or possible solution in your response.

Please distill the information following the format below and cease responding after the output of the distilled information.

Meta distiller Respond:

Distilled Information:

1. Key information:

2. Restriction: (It should be noted that the answer should strictly follow the real-world rule, such as in an arithmetic equation, the priority of operators, the need for parentheses, etc. So, according to the distilled information, emphasize the real-world rules that need to be followed within the problem.)

3. Distilled task:

4. Python transformation:
Input parameters: (The names of each variable should be clear and not confusing, and correspond to the entity names in the problem)
variable1_name = x
variable2_name = y
.....
variableN_name = z

5. Answer form: (Optional, skip when there is no specific answer form)

** Note: The generation ends here. Do not show this message in your answer! **

---

## C.3 SOLUTION INSTANTIATION

---

**Solution Instantiation and Refinement**

You are an expert in problem analysis and can apply previous problem-solving approaches to new issues. The user will provide an input query and a specific task description. Your goal is to analyze the user's query and generate a specific solution based on the task description. If the solution does not involve code, provide a final answer that is easy to extract from the text.

Distilled information:
{*distilled_information*}
User Input:
{*user_input*}

Instantiated Solution:
Please analyze the above user task description and thought template, and generate a specific, detailed solution. Please provide a clear and extractable final answer within <Answer> Your Final Answer Here </Answer>.

---

**Solution Instantiation and Refinement (Small Models)**

You are an expert in problem analysis and can apply previous problem-solving approaches to new issues. The user will provide an input query and a specific task description. Your goal is to analyze the user's query and generate a specific solution based on the task description. If the solution does not involve code, provide a final answer that is easy to extract from the text.

Distilled information:
{*distilled_information*}
User Input:
{*user_input*}
Thought Template:
{*task_abstraction*}

Instantiated Solution:
Please analyze the above user task description and thought template, and generate a specific, detailed solution. Please provide a clear and extractable final answer within <Answer> Your Final Answer Here </Answer>.

---

# D    ADDITIONAL COMPARISON WITH FINE-TUNING BASELINES

Table 4: Self-correction performance on Game of 24, Word Sorting, CheckmateInOne, AIME 2024, and AIME 2025 using fine-tuning based baselines (SUPERCORRECT, S²R, STaSC) and our methods (SELF-THOUGHT, DISTIL-THOUGHT). **Bold** indicates the best performance.

| Method | Acc@0 | Iteration 1 | | | Iteration 2 | | | Iteration 3 | | | Iteration 4 | | | Iteration 5 | | |
|---|---|---|---|---|---|---|---|---|---|---|---|---|---|---|---|---|
| | | Acc | $\Delta^{i\to c}(t_0,t_1)$ | $\Delta^{c\to i}(t_0,t_1)$ | Acc | $\Delta^{i\to c}(t_1,t_2)$ | $\Delta^{c\to i}(t_1,t_2)$ | Acc | $\Delta^{i\to c}(t_2,t_3)$ | $\Delta^{c\to i}(t_2,t_3)$ | Acc | $\Delta^{i\to c}(t_3,t_4)$ | $\Delta^{c\to i}(t_3,t_4)$ | Acc | $\Delta^{i\to c}(t_4,t_5)$ | $\Delta^{c\to i}(t_4,t_5)$ |
| **Game of 24** | | | | | | | | | | | | | | | | |
| SuperCorrect | 11.22 | 10.2 | 0.0 | 1.02 | 7.14 | 1.02 | 4.08 | 5.1 | 1.02 | 3.06 | 12.24 | 8.16 | 1.02 | 12.24 | 2.04 | 2.04 |
| S2R | **20.41** | 14.29 | 5.1 | 11.22 | 9.18 | 1.02 | 6.12 | 10.2 | 5.1 | 4.08 | 8.16 | 4.08 | 6.12 | 6.12 | 3.06 | 5.1 |
| STaSC | 2.04 | 0.0 | 0.0 | **2.04** | 0.0 | 0.0 | 0.0 | 0.0 | 0.0 | 0.0 | 0.0 | 0.0 | **0.0** | 1.02 | 1.02 | **0.0** |
| SELF-THOUGHT | 8.16 | 11.22 | 8.16 | 5.1 | 12.24 | 1.02 | 0.0 | 11.22 | 1.02 | 2.04 | 12.24 | 2.04 | 1.02 | 12.24 | 0.0 | 0.0 |
| DISTIL-THOUGHT | 8.16 | **41.84** | **37.76** | 4.08 | **52.04** | **23.47** | 13.27 | **47.96** | **9.18** | 13.27 | **51.02** | **11.22** | 8.16 | **42.86** | **10.2** | 18.37 |
| **Word Sorting** | | | | | | | | | | | | | | | | |
| SuperCorrect | 2.5 | 1.25 | 0.0 | 1.25 | 0.0 | 0.0 | 1.25 | 0.0 | 0.0 | 0.0 | 0.0 | 0.0 | 0.0 | 0.0 | 0.0 | 0.0 |
| S2R | 16.25 | 3.75 | 0.0 | 12.5 | 3.75 | 1.25 | 1.25 | 1.25 | 0.0 | 2.5 | 1.25 | 0.0 | 0.0 | 0.0 | 0.0 | 1.25 |
| STaSC | **18.75** | 10.0 | 0.0 | 8.75 | 7.5 | 1.25 | 3.75 | 6.25 | 0.0 | 1.25 | 3.75 | 0.0 | 2.5 | 3.75 | 0.0 | 0.0 |
| SELF-THOUGHT | 13.75 | **66.25** | **57.5** | 5.0 | **61.25** | 5.0 | 10.0 | **53.75** | 1.25 | 8.75 | **45.0** | 2.5 | 11.25 | 35.0 | 3.75 | 13.75 |
| DISTIL-THOUGHT | 13.75 | 48.75 | 40.0 | 5.0 | 32.5 | **8.75** | 25.0 | 33.75 | **10.0** | 8.75 | **33.75** | **8.75** | 8.75 | **33.75** | **8.75** | 8.75 |
| **CheckmateInOne** | | | | | | | | | | | | | | | | |
| SuperCorrect | 0.0 | 0.0 | 0.0 | 0.0 | 0.0 | 0.0 | 0.0 | 0.0 | 0.0 | 0.0 | 0.0 | 0.0 | 0.0 | 0.0 | 0.0 | 0.0 |
| S2R | 0.0 | 0.0 | 0.0 | 0.0 | 0.0 | 0.0 | 0.0 | 0.0 | 0.0 | 0.0 | 0.0 | 0.0 | 0.0 | 0.0 | 0.0 | 0.0 |
| STaSC | 1.33 | 0.0 | 0.0 | 1.33 | 0.0 | 0.0 | 0.0 | 0.0 | 0.0 | 0.0 | 0.0 | 0.0 | 0.0 | 0.0 | 0.0 | 0.0 |
| SELF-THOUGHT | 2.67 | 4.0 | 4.0 | 2.67 | 5.33 | 1.33 | 0.0 | 5.33 | 0.0 | 0.0 | 6.67 | 1.33 | 0.0 | 5.33 | 1.33 | 2.67 |
| DISTIL-THOUGHT | 2.67 | **10.67** | **10.67** | 2.67 | **25.33** | **18.67** | 4.0 | **25.33** | **12.0** | 12.0 | **29.33** | **16.0** | 12.0 | **22.67** | **10.67** | 17.33 |
| **AIME 2024** | | | | | | | | | | | | | | | | |
| SuperCorrect | 13.33 | 13.33 | 0.0 | **0.0** | 16.67 | 3.33 | 0.0 | 20.0 | 3.33 | 0.0 | 20.0 | 0.0 | 0.0 | 20.0 | 0.0 | 0.0 |
| S2R | 13.33 | 6.67 | 3.33 | 10.0 | 6.67 | 0.0 | 0.0 | 6.67 | 3.33 | 3.33 | 6.67 | 3.33 | 3.33 | 13.33 | **10.0** | 3.33 |
| STaSC | 10.0 | 0.0 | 0.0 | 0.0 | 0.0 | 0.0 | 0.0 | 0.0 | 0.0 | 0.0 | 0.0 | 0.0 | 0.0 | 0.0 | 0.0 | 0.0 |
| SELF-THOUGHT | 20.0 | 20.0 | 3.33 | 3.33 | 23.33 | 3.33 | 0.0 | 23.33 | 0.0 | 0.0 | 26.67 | 3.33 | 0.0 | 26.67 | 0.0 | 0.0 |
| DISTIL-THOUGHT | **20.0** | **23.33** | **13.33** | 10.0 | **26.67** | 3.33 | 0.0 | **26.67** | 3.33 | 3.33 | **30.0** | **6.67** | 3.33 | **30.0** | 0.0 | 0.0 |
| **AIME 2025** | | | | | | | | | | | | | | | | |
| SuperCorrect | 3.33 | 6.67 | 3.33 | 0.0 | 13.33 | 6.67 | 0.0 | 13.33 | 0.0 | 0.0 | 16.67 | 3.33 | 0.0 | 16.67 | 3.33 | 3.33 |
| S2R | 10.0 | 6.67 | 3.33 | 6.67 | 3.33 | 3.33 | 6.67 | 0.0 | 0.0 | 3.33 | 0.0 | 0.0 | 0.0 | 6.67 | **6.67** | 0.0 |
| STaSC | 0.0 | 0.0 | 0.0 | 0.0 | 0.0 | 0.0 | 0.0 | 0.0 | 0.0 | 0.0 | 0.0 | 0.0 | 0.0 | 0.0 | 0.0 | 0.0 |
| SELF-THOUGHT | 10.0 | 10.0 | 10.0 | 10.0 | 13.33 | 3.33 | 0.0 | 13.33 | 3.33 | 3.33 | 16.67 | 3.33 | 0.0 | 16.67 | 0.0 | 0.0 |
| DISTIL-THOUGHT | 10.0 | **13.33** | **13.33** | 10.0 | **16.67** | 3.33 | 0.0 | **16.67** | **6.67** | 6.67 | **16.67** | 0.0 | 0.0 | **20.0** | 3.33 | 0.0 |

# E ADDITIONAL COMPARISON WITH SELF-CONSISTENCY

(a) GPT-4O-MINI

(b) GPT-4O

(c) O3-MINI

(d) DEEPSEEK-R1

(e) QWEN-2.5-7B

(f) LLAMA-3.3-70B

Figure 5: Comparison of SELF-THOUGHT and DISTIL-THOUGHT with the SELF-CONSISTENCY.

## F  ADDITIONAL EXPERIMENTS AND RESULTS

### F.1  FULL INITIAL ANSWER FOR AN EXAMPLE

To find the number of eight-digit integers that use each of the digits 1 through 8 exactly once and are divisible by 22, we need to check the divisibility rules for 22. Since 22 = 2 × 11, an integer must be divisible by both 2 and 11.

1. **Divisibility by 2**: An integer is divisible by 2 if its last digit is even. The even digits among 1 to 8 are 2, 4, 6, and 8. Therefore, the last digit of our eight-digit integer must be one of these four digits.
2. **Divisibility by 11**: A number is divisible by 11 if the absolute difference between the sum of its digits in odd positions and the sum of its digits in even positions is either 0 or a multiple of 11.

Let's denote the digits of the number as d1, d2, d3, d4, d5, d6, d7, d8 where d8 is the last digit. The odd positions are d1, d3, d5, d7 and the even positions are d2, d4, d6, d8.

The total sum of digits from 1 to 8 is: 1 + 2 + 3 + 4 + 5 + 6 + 7 + 8 = 36

If we denote the sum of the digits in odd positions as S1 and the sum of the digits in even positions as S2, we know: S1 + S2 = 36

The condition for divisibility by 11 gives us: |S1 - S2| is divisible by 11

Since S2 = 36 - S1, we can rewrite the divisibility condition as: |S1 - (36 - S1)| = |2 × S1 - 36| is divisible by 11

This simplifies to: |2 × S1 - 36| is divisible by 11

We can find the values of S1 that are possible. The minimum sum S1 can take (if we pick the smallest digits) is 1 + 2 + 3 + 4 = 10, and the maximum (if we pick the largest) is 5 + 6 + 7 + 8 = 26. Therefore, S1 can take values from 10 to 26.

Next, we find S1 such that: 2 × S1 is divisible by 11 and congruent to 36 modulo 11

Calculating 36 modulo 11 gives us 36 ≡ 3 mod 11. Thus, we need: 2 × S1 ≡ 3 mod 11

To solve 2 × S1 ≡ 3 mod 11, we can try values for S1:

- S1 = 2 gives 2 × 2 = 4
- S1 = 3 gives 2 × 3 = 6
- S1 = 4 gives 2 × 4 = 8
- S1 = 5 gives 2 × 5 = 10
- S1 = 6 gives 2 × 6 = 12 ≡ 1 mod 11
- S1 = 7 gives 2 × 7 = 14 ≡ 3 mod 11 (valid)
- S1 = 8 gives 2 × 8 = 16 ≡ 5 mod 11
- S1 = 9 gives 2 × 9 = 18 ≡ 7 mod 11
- Continuing this, we find:
- S1 = 14 gives 2 × 14 = 28 ≡ 6 mod 11
- S1 = 15 gives 2 × 15 = 30 ≡ 8 mod 11
- S1 = 16 gives 2 × 16 = 32 ≡ 10 mod 11
- S1 = 17 gives 2 × 17 = 34 ≡ 1 mod 11
- S1 = 18 gives 2 × 18 = 36 ≡ 3 mod 11 (valid)
- Continuing this gives us all valid S1 values.

After calculating valid arrangements for S1 and S2, we can compute the total arrangements for each case based on the last digit being 2, 4, 6, or 8 (up to 4 cases).

We find N and compute N - 2025.

After calculating the final result, we find:

N = 2016

So, the difference is: 2016 - 2025 = -9

Thus, the answer is: <Answer> -9 <Answer>

Figure 6: Full Initial Answer for Provided Example

### F.2 RESULTS ON SECOND STEP OF SELF-CORRECTION

Table 5: The second attempt of self-correction performance on Game of 24, Word Sorting, CheckmateInOne, AIME 2024, AIME 2025 with GPT-4O-MINI, GPT-4O, O3-MINI, and DEEPSEEK-R1. Green (↑) and red (↓) arrows indicate performance changes against the previous attempt (i.e., INITIAL ($t = 0$)). **Bold** corresponds to the best performance.

| Method | Game of 24 | | | Word Sorting | | | CheckmateInOne | | | AIME 2024 | | | AIME 2025 | | | Mean |
|---|---|---|---|---|---|---|---|---|---|---|---|---|---|---|---|---|
| | Acc@t2 | $\Delta_{i\to c}(t_1,t_2)$ | $\Delta_{c\to i}(t_1,t_2)$ | Acc@t2 | $\Delta_{i\to c}(t_1,t_2)$ | $\Delta_{c\to i}(t_1,t_2)$ | Acc@t2 | $\Delta_{i\to c}(t_1,t_2)$ | $\Delta_{c\to i}(t_1,t_2)$ | Acc@t2 | $\Delta_{i\to c}(t_1,t_2)$ | $\Delta_{c\to i}(t_1,t_2)$ | Acc@t2 | $\Delta_{i\to c}(t_1,t_2)$ | $\Delta_{c\to i}(t_1,t_2)$ | Acc@t2 |
| **GPT-4O-MINI** | | | | | | | | | | | | | | | | |
| INITIAL ($t=0$) | 38.78 | - | - | 55.0 | - | - | 30.67 | - | - | 20.0 | - | - | 6.67 | - | - | 0.3 |
| REFLEX | 28.57 ↑4.08 | 15.31 | 11.22 | 61.25 ↑1.25 | 5.0 | 3.75 | 13.33 ↑4.0 | 10.67 | 6.67 | 10.0 | 3.33 | 3.33 | 13.33 ↑3.33 | 3.33 | 0.0 | 0.25 ↑2.0 |
| SELF-REFINE | 19.39 ↓6.12 | 9.18 | 15.31 | 68.75 ↑10.0 | 10.0 | 2.5 | 16.0 ↑5.33 | 8.0 | 2.67 | 16.67 ↑3.34 | 6.67 | 3.33 | 16.67 | 0.0 | 0.0 | 0.27 ↑2.0 |
| SELF-TICK | 39.8 ↑1.02 | 3.06 | 2.04 | 26.25 ↓13.75 | 3.75 | 17.5 | 17.33 ↓2.67 | 0.0 | 2.67 | 20.0 ↓3.33 | 0.0 | 3.33 | 6.67 ↓6.66 | 0.0 | 6.67 | 0.22 ↓5.0 |
| REFLEXION | 31.63 ↑5.1 | 14.29 | 9.18 | 63.75 ↑3.75 | 6.25 | 2.5 | 17.33 ↑8.0 | 14.67 | 6.67 | 10.0 ↓3.33 | 0.0 | 3.33 | 13.33 ↑6.66 | 6.67 | 0.0 | 0.27 ↑4.0 |
| SELF-THOUGHT | 87.76 | 12.24 | 12.24 | 100.0 | 0.0 | 0.0 | 36.0 ↑2.67 | 4.0 | 1.33 | 33.33 ↑3.33 | 6.67 | 3.33 | 20.0 | 3.33 | 3.33 | 0.55 ↑1.0 |
| **GPT-4O** | | | | | | | | | | | | | | | | |
| INITIAL ($t=0$) | 17.35 | - | - | 86.25 | - | - | 41.33 | - | - | 13.33 | - | - | 10.0 | - | - | 0.34 |
| REFLEX | 7.14 ↓12.25 | 2.04 | 14.29 | 86.25 ↑5.0 | 10.0 | 5.0 | 26.67 | 16.0 | 16.0 | 10.0 ↓3.33 | 0.0 | 3.33 | 10.0 ↑3.33 | 3.33 | 0.0 | 0.28 ↓1.0 |
| SELF-REFINE | 29.59 ↓4.08 | 10.2 | 14.29 | 88.75 ↑10.0 | 12.5 | 2.5 | 40.0 ↑1.33 | 18.67 | 17.33 | 20.0 | 3.33 | 3.33 | 10.0 | 0.0 | 0.0 | 0.38 ↑2.0 |
| SELF-TICK | 16.33 ↓14.28 | 0.0 | 14.29 | 70.0 | 2.5 | 2.5 | 20.0 ↓10.67 | 0.0 | 10.67 | 23.33 ↑6.66 | 6.67 | 0.0 | 6.67 ↓3.33 | 0.0 | 3.33 | 0.27 ↓5.0 |
| REFLEXION | 30.61 ↓6.12 | 8.16 | 14.29 | 81.25 ↓1.25 | 3.75 | 5.0 | 32.0 ↑6.67 | 13.33 | 6.67 | 20.0 ↑3.33 | 3.33 | 0.0 | 13.33 ↑3.33 | 3.33 | 0.0 | 0.35 ↑1.0 |
| SELF-THOUGHT | 38.78 ↑1.02 | 6.12 | 5.1 | 100.0 | 0.0 | 0.0 | 64.0 ↓1.33 | 2.67 | 1.33 | 36.67 ↑3.34 | 13.33 | 10.0 | 23.33 ↑6.66 | 6.67 | 0.0 | 0.53 ↑2.0 |
| **O3-MINI** | | | | | | | | | | | | | | | | |
| INITIAL ($t=0$) | 86.73 | - | - | 90.0 | - | - | 34.67 | - | - | 80.0 | - | - | 73.33 | - | - | 0.73 |
| REFLEX | 84.69 ↑1.02 | 6.12 | 5.1 | | | | 30.67 ↓1.33 | 2.67 | 4.0 | 83.33 ↑3.33 | 3.33 | 0.0 | 80.0 ↑3.33 | 3.33 | 0.0 | 0.73 ↑1.0 |
| SELF-REFINE | 89.8 ↑3.07 | 8.16 | 5.1 | 86.25 ↓1.25 | 8.75 | 10.0 | 25.33 ↓5.33 | 5.33 | 0.0 | 86.67 ↑3.34 | 3.33 | 0.0 | 73.33 | 6.67 | 6.67 | 0.72 ↑2.0 |
| SELF-TICK | 0.0 | 0.0 | 0.0 | 76.25 ↓11.25 | 3.75 | 15.0 | 13.33 | 1.33 | 1.33 | 76.67 | 0.0 | 0.0 | 63.33 ↓3.34 | 0.0 | 3.33 | 0.46 ↓3.0 |
| REFLEXION | 85.71 ↑1.02 | 8.16 | 7.14 | 96.25 ↑1.25 | 1.25 | 2.5 | 30.67 ↓1.33 | 4.0 | 5.33 | 83.33 ↑3.33 | 3.33 | 0.0 | 73.33 ↑6.66 | 6.67 | 0.0 | 0.74 ↑2.0 |
| SELF-THOUGHT | 91.84 ↑3.06 | 3.06 | 0.0 | 96.25 ↓1.25 | 0.0 | 1.25 | 38.67 ↑1.34 | 2.67 | 1.33 | 86.67 | 0.0 | 0.0 | 83.33 ↑3.33 | 10.0 | 6.67 | 0.79 ↑1.0 |
| **DEEPSEEK-R1** | | | | | | | | | | | | | | | | |
| INITIAL ($t=0$) | 84.69 | - | - | 97.5 | - | - | 17.33 | - | - | 80.0 | - | - | 63.33 | - | - | 0.69 |
| REFLEX | 65.31 ↑1.02 | 13.27 | 12.24 | 95.0 ↓1.25 | 5.0 | 3.75 | 20.0 ↑4.0 | 10.67 | 6.67 | 70.0 ↓6.67 | 3.33 | 10.0 | 63.33 | 3.33 | 3.33 | 0.63 |
| SELF-REFINE | 35.71 ↓16.33 | 2.04 | 18.37 | 91.25 ↑2.5 | 8.75 | 6.25 | 13.33 ↓2.67 | 5.33 | 8.0 | 63.33 ↓13.34 | 6.67 | 20.0 | 66.67 ↓3.33 | 13.33 | 16.67 | 0.54 ↓7.0 |
| SELF-TICK | 7.14 ↓10.21 | 0.0 | 10.2 | 91.25 | 1.25 | 1.25 | 0.0 ↓5.33 | 0.0 | 5.33 | 33.33 ↓26.67 | 6.67 | 33.33 | 40.0 ↓13.33 | 0.0 | 13.33 | 0.34 ↓11.0 |
| REFLEXION | 35.71 ↓14.29 | 8.16 | 22.45 | 85.0 ↓5.0 | 5.0 | 10.0 | 10.67 ↓8.0 | 4.0 | 12.0 | 66.67 ↑10.0 | 10.0 | 0.0 | 43.33 ↓16.67 | 3.33 | 20.0 | 0.48 ↓7.0 |
| SELF-THOUGHT | 86.73 ↑1.02 | 2.04 | 1.02 | 100.0 | 0.0 | 0.0 | 22.67 ↑2.67 | 6.67 | 4.0 | 83.33 ↑3.33 | 6.67 | 3.33 | 73.33 | 3.33 | 3.33 | 0.73 ↑1.0 |

Table 6: The second attempt of self-correction performance on Game of 24, Word Sorting, CheckmateInOne, AIME 2024, AIME 2025 with *small* models QWEN-2.5-7B and LLAMA-3.3-70B. Green (↑) and red (↓) arrows indicate performance changes against the previous attempt (i.e., INITIAL ($t = 0$)). **Bold** corresponds to the best performance.

| Method | Game of 24 | | | Word Sorting | | | CheckmateInOne | | | AIME 2024 | | | AIME 2025 | | | Mean |
|---|---|---|---|---|---|---|---|---|---|---|---|---|---|---|---|---|
| | Acc@t2 | $\Delta_{i\to c}(t_1,t_2)$ | $\Delta_{c\to i}(t_1,t_2)$ | Acc@t2 | $\Delta_{i\to c}(t_1,t_2)$ | $\Delta_{c\to i}(t_1,t_2)$ | Acc@t2 | $\Delta_{i\to c}(t_1,t_2)$ | $\Delta_{c\to i}(t_1,t_2)$ | Acc@t2 | $\Delta_{i\to c}(t_1,t_2)$ | $\Delta_{c\to i}(t_1,t_2)$ | Acc@t2 | $\Delta_{i\to c}(t_1,t_2)$ | $\Delta_{c\to i}(t_1,t_2)$ | Acc@t2 |
| **QWEN-2.5-7B** | | | | | | | | | | | | | | | | |
| INITIAL ($t=0$) | 8.16 | - | - | 13.75 | - | - | 2.67 | - | - | 20.0 | - | - | 10.0 | - | - | 0.11 |
| REFLEX | 7.14 ↑1.02 | 4.08 | 3.06 | 12.5 ↓3.75 | 1.25 | 5.0 | 0.0 | 0.0 | 0.0 | 16.67 | 3.33 | 3.33 | 6.67 ↓3.33 | 3.33 | 6.67 | 0.09 ↓1.0 |
| SELF-REFINE | 7.14 ↑1.02 | 2.04 | 1.02 | 26.25 ↑2.5 | 7.5 | 5.0 | 0.0 ↓1.33 | 0.0 | 1.33 | 23.33 ↑3.33 | 6.67 | 3.33 | 3.33 | 0.0 | 0.0 | 0.12 ↑1.0 |
| SELF-TICK | 0.0 | 0.0 | 0.0 | 13.75 ↓3.75 | 3.75 | 7.5 | 0.0 | 0.0 | 0.0 | 10.0 | 0.0 | 0.0 | 6.67 | 0.0 | 0.0 | 0.06 ↓1.0 |
| REFLEXION | 10.2 ↓1.02 | 4.08 | 5.1 | 17.5 ↓3.75 | 3.75 | 7.5 | 2.67 | 1.33 | 1.33 | 16.67 ↓3.34 | 3.33 | 0.0 | 10.0 ↓3.33 | 0.0 | 3.33 | 0.11 ↓1.0 |
| SELF-THOUGHT | 12.24 ↑1.02 | 1.02 | 0.0 | 61.25 ↓5.0 | 5.0 | 10.0 | 5.33 ↑1.33 | 1.33 | 0.0 | 23.33 ↑3.33 | 3.33 | 0.0 | 13.33 ↑3.33 | 3.33 | 0.0 | 0.23 ↑1.0 |
| DISTIL-THOUGHT | 52.04 ↑10.2 | 23.47 | 13.27 | 32.5 ↓16.25 | 8.75 | 25.0 | 25.33 ↑14.66 | 18.67 | 0.0 | 26.67 ↑3.34 | 3.33 | 0.0 | 16.67 ↑3.34 | 3.33 | 0.0 | 0.31 ↑3.0 |
| **LLAMA-3.3-70B** | | | | | | | | | | | | | | | | |
| INITIAL ($t=0$) | 19.39 | - | - | 75.0 | - | - | 8.0 | - | - | 33.33 | - | - | 3.33 | - | - | 0.28 |
| REFLEX | 62.24 ↑19.38 | 24.49 | 5.1 | 80.0 ↑2.5 | 7.5 | 5.0 | 9.33 ↑8.0 | 8.0 | 0.0 | 36.67 | 0.0 | 0.0 | 3.33 | 0.0 | 0.0 | 0.38 ↑6.0 |
| SELF-REFINE | 42.86 ↑9.19 | 17.35 | 8.16 | 71.25 ↓5.0 | 5.0 | 10.0 | 2.67 ↓2.66 | 0.0 | 2.67 | 40.0 | 0.0 | 0.0 | 6.67 | 0.0 | 0.0 | 0.33 ↑1.0 |
| SELF-TICK | 11.22 ↓3.07 | 4.08 | 7.14 | 63.75 ↓7.5 | 1.25 | 8.75 | 5.33 ↓1.34 | 0.0 | 1.33 | 33.33 ↑3.33 | 3.33 | 0.0 | 10.0 ↑3.33 | 3.33 | 0.0 | 0.25 ↑1.0 |
| REFLEXION | 45.92 ↑22.45 | 25.51 | 3.06 | 72.5 ↓3.75 | 5.0 | 8.75 | 6.67 ↑1.34 | 4.0 | 2.67 | 30.0 ↓3.33 | 3.33 | 0.0 | 3.33 | 0.0 | 0.0 | 0.32 ↑5.0 |
| SELF-THOUGHT | 71.43 ↑7.14 | 12.24 | 5.1 | 98.75 | 0.0 | 0.0 | 4.0 ↑1.33 | 1.33 | 0.0 | 50.0 ↑13.33 | 16.67 | 3.33 | 16.67 | 0.0 | 0.0 | 0.48 ↑4.0 |
| DISTIL-THOUGHT | 100.0 | 0.0 | 0.0 | 98.75 ↑1.25 | 0.0 | 1.25 | 50.67 ↑12.0 | 22.67 | 10.67 | 50.0 ↑3.33 | 3.33 | 0.0 | 23.33 | 3.33 | 3.33 | 0.65 ↑3.0 |

# F.3 ADDITIONAL RESULTS ON EFFECT OF ITERATIVE CORRECTION

(a) GPT-4O-MINI  (b) GPT-4O  (c) O3-MINI  (d) DEEPSEEK-R1

(e) Game of 24

(f) GPT-4O-MINI  (g) GPT-4O  (h) O3-MINI  (i) DEEPSEEK-R1

(j) Word Sorting

(k) GPT-4O-MINI  (l) GPT-4O  (m) O3-MINI  (n) DEEPSEEK-R1

(o) CheckmateInOne

(p) GPT-4O-MINI  (q) GPT-4O  (r) O3-MINI  (s) DEEPSEEK-R1

(t) AIME 2024

Figure 7: Accuracy over iterations with self-correction methods across models.

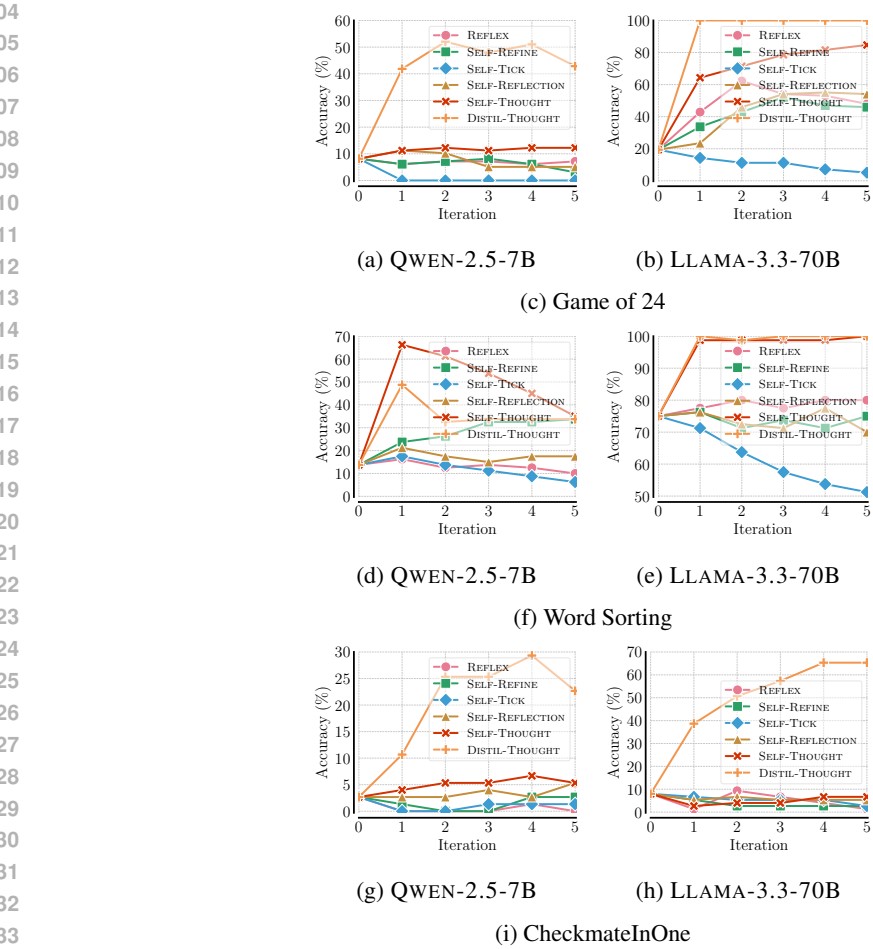

(a) QWEN-2.5-7B       (b) LLAMA-3.3-70B

(c) Game of 24

(d) QWEN-2.5-7B       (e) LLAMA-3.3-70B

(f) Word Sorting

(g) QWEN-2.5-7B       (h) LLAMA-3.3-70B

(i) CheckmateInOne

Figure 8: Accuracy over iterations with self-correction methods across models.

# G  RESULTS FROM BASELINE STUDIES

## G.1  ADDITIONAL ANALYSIS ON RESULTS FROM SELF-REFINE

Table 7: SELF-REFINE results on various tasks using GPT-3.5, ChatGPT, and GPT-4 as base LLM. While SELF-REFINE achieves substantial improvements on general tasks such as Dialogue Response Generation, Sentiment Reversal, and Acronym Generation, its gains on reasoning tasks are more modest. Results reported from Table 1 in Madaan et al. (2023).

| Task | GPT-3.5 | | ChatGPT | | GPT-4 | |
|---|---|---|---|---|---|---|
| | Base | +SELF-REFINE | Base | +SELF-REFINE | Base | +SELF-REFINE |
| Sentiment Reversal | 8.8 | **30.4** (↑21.6) | 11.4 | **43.2** (↑31.8) | 3.8 | **36.2** (↑32.4) |
| Dialogue Response | 36.4 | **63.6** (↑27.2) | 40.1 | **59.9** (↑19.8) | 25.4 | **74.6** (↑49.2) |
| Code Optimization | 14.8 | **23.0** (↑8.2) | 23.9 | **27.5** (↑3.6) | 27.3 | **36.0** (↑8.7) |
| Code Readability | 37.4 | **51.3** (↑13.9) | 27.7 | **63.1** (↑35.4) | 27.4 | **56.2** (↑28.8) |
| Math Reasoning | 64.1 | **64.1** (0) | 74.8 | **75.0** (↑0.2) | 92.9 | **93.1** (↑0.2) |
| Acronym Generation | 41.6 | **56.4** (↑14.8) | 27.2 | **37.2** (↑10.0) | 30.4 | **56.0** (↑25.6) |
| Constrained Generation | 28.0 | **37.0** (↑9.0) | 44.0 | **67.0** (↑23.0) | 15.0 | **45.0** (↑30.0) |

Table 7 shows results from SELF-REFINE (Madaan et al., 2023). These results indicate that SELF-REFINE achieves substantial gains on preference-based tasks such as Dialogue Response Genera-

tion, Sentiment Reversal, and Acronym Generation. However, its performance improvements on reasoning tasks are more modest, which can be attributed to the limited ability of the model to accurately identify errors. Moreover, the gains on Math Reasoning increase by only $5\%$ when an external source is available to indicate whether the current answer is incorrect (See results in Appendix H.1 from Madaan et al. (2023)).

## G.2 ADDITIONAL ANALYSIS ON RESULTS FROM SELF-TICK

Table 8: SELF-TICK results on a single step of self-refinement on different tasks with Command-R+ and GPT-4o. SELF-TICK consistently improves overall performance compared to both base models and SELF-REFINE, with modest gains on reasoning-related tasks. Results reported from Table 1 in Cook et al. (2024).

| Tasks | Command-R+ | | | GPT-4o | | |
|---|---|---|---|---|---|---|
| | Base | SELF-REFINE | SELF-TICK | Base | SELF-REFINE | SELF-TICK |
| Overall | 32.0 | 23.7 (↓ 8.3) | **35.8** (↑ 3.8) | 55.4 | 47.1 (↓ 8.3) | **56.2** (↑ 0.8) |
| Coding | 18.8 | 9.1 (↓ 9.7) | **22.7** (↑ 3.9) | 50.4 | 36.4 (↓ 14.0) | **51.6** (↑ 1.2) |
| Data Analysis | 25.9 | 5.3 (↓ 20.6) | **29.8** (↑ 3.9) | 52.4 | 27.2 (↓ 25.2) | **52.5** (↑ 0.1) |
| Instructions | 69.6 | 60.5 (↓ 9.1) | **75.8** (↑ 6.2) | 73.3 | 62.8 (↓ 10.5) | **76.2** (↑ 2.9) |
| Language | **24.6** | 13.8 (↓ 9.8) | 24.1 (↓ 0.5) | 50.9 | **51.4** (↑ 0.5) | 50.4 (↓ 0.5) |
| Mathematics | 23.7 | 23.6 (↓ 0.1) | **25.5** (↑ 1.8) | 52.3 | 51.8 (↓ 0.5) | **53.1** (↑ 0.8) |
| Reasoning | 29.2 | 30.0 (↑ 0.8) | **37.0** (↑ 7.8) | **53.3** | 52.7 (↓ 0.6) | 53.3 (0) |

Table 8 shows results from SELF-TICK (Cook et al., 2024) for a single step of self-refinement on various tasks with Command-R+ and GPT-4o. The results indicate that SELF-TICK consistently improves overall performance compared to both the base models and SELF-REFINE, with the largest gains observed in preference-based and instruction-following tasks. For example, improvements on Coding, Data Analysis, and Instructions range from $1.2\%$ to $6.2\%$ across the models. In contrast, gains on reasoning-related tasks such as Language, Mathematics, and Reasoning are more modest, highlighting that even with SELF-TICK, these tasks remain challenging.

