# OpenReview forum: "Self-Correction via Task Distillation"
_ICLR.cc/2026/Conference — ICLR 2026 Conference Withdrawn Submission_

### Official Review · Reviewer_TaA6 · 2025-10-31

**Soundness:** 3
**Presentation:** 3
**Contribution:** 2
**Rating:** 4
**Confidence:** 3

**Summary:**

The paper introduces a framework for LLM self-correction, where the main idea is to generate an intermediate task abstraction before refining the answer. This is opposed to the vanilla setup that critiques the output directly, the propose approach distills the problem and output into a structured template that encodes variables, constraints and other information. This abstraction is then used to generate a better solution. The authors also propose to borrow abstractions generated from more capable models and use them with smaller model to improve their self-correction ability. Evaluation lacks deeper analysis
While results are impressive, the paper treats accuracy as the main measure of success. There is little qualitative or error-type analysis showing what kinds of reasoning errors SELF-THOUGHT actually fixes or fails to fix.Across reasoning benchmarks such as AIME 2024/2025, game of 24, CheckmateInOne, and word sorting, the method shows empirical gains and outperforms baselines such Self-Refine and Reflexion.

**Strengths:**

1. The proposed approach is simple and straightforward, does require training, and can be applied to different models.
2. The experiments are comprehensive and Self-Thought shows gains compared to the baselines on different reasoning tasks.

**Weaknesses:**

1. **Lack of technical novelty**: The main contribution is the task abstraction step, which basically structures the relevant elements to the problem into a template conducive for self-correction. I fail to see any technical depth to the paper. Also, the approach seems to be mostly helpful with smaller models that require some hand-holding via task abstraction. I suspect this approach will help larger and more capable models.

2. **lack of deeper analysis** While results are impressive, the paper treats accuracy as the main measure of success. There is little qualitative or error-type analysis showing what kinds of reasoning errors Self-Thought actually fixes or fails to fix. It also remains unclear what types of abstractions are useful or harmful.

3. **Distill-Thought is impractical**:  Why sample a task abstraction from a large model then use it with a small model? If we have access to a large, mode capable model, why not use it right away?

4. The authors randomly samples abstractions from successful gpt-4o-mini runs. This feels arbitrary. Without an analysis of abstraction quality or diversity, it’s unclear how reproducible or stable the improvements are. In addition, the approach does not seem to benefit from more iterations and tends to plateau early.

5. This is minor but the figures and tables are dense and visually repetitive. A compact summary plot (accuracy vs model size vs method) could make the takeaways more immediate.

**Questions:**

See weaknesses

---

### Official Review · Reviewer_ZJkE · 2025-10-31

**Soundness:** 2
**Presentation:** 2
**Contribution:** 2
**Rating:** 2
**Confidence:** 4

**Summary:**

This paper introduces SELF-THOUGHT, a novel self-correction framework that improves LLM reasoning through task abstraction before solution refinement. Instead of directly critiquing outputs, the model first distills problems into structured templates capturing key variables and constraints, then uses these abstractions to guide more robust corrections. The key innovation is DISTIL-THOUGHT, which transfers task abstractions from larger models to smaller ones, enabling effective self-correction in resource-constrained settings without expensive fine-tuning. Experiments across reasoning benchmarks show substantial improvements.

**Strengths:**

1. The paper effectively reframes self-correction as a task understanding problem rather than surface-level output refinement. It identifies a critical gap in existing methods, which fail to address reasoning flaws or benefit smaller models.
2. The paper presents extensive experiments across diverse model sizes and reasoning-intensive benchmarks, showing consistent and substantial improvements over multiple baselines.

**Weaknesses:**

1. Overreliance on prompt engineering without fundamental innovation. The proposed method primarily relies on prompt engineering rather than introducing a principled algorithmic or architectural improvement. The “task abstraction” stage is essentially another prompted generation step, which can be error-prone and propagate mistakes into subsequent reasoning. Moreover, the paper lacks rigorous analysis of abstraction failures and their impact on final task performance, leaving the method’s robustness and reliability unclear.

2. Lack of principled guidance for constructing task abstractions.The paper provides no systematic methodology for designing effective task abstractions, instead relying on task-specific prompt templates. There are no formal criteria or independent metrics to assess abstraction quality, making it impossible to separate abstraction effectiveness from overall task performance. Moreover, the paper does not compare alternative abstraction strategies, leaving unclear whether the reported gains stem from the abstraction concept itself or from ad-hoc template choices.

3. Although the paper includes several prompt-based self-correction baselines in the main text and training-based methods in the appendix, it omits comparisons and discussion with the most conceptually related work, such as [1]. Without evaluating against these closely aligned works, it is difficult to assess the true novelty and relative advantage of the proposed method.

4. The proposed task abstraction approach appears most suitable for problems with well-defined correct answers and objective evaluation criteria. All evaluated benchmarks involve mathematical reasoning or logical puzzles where structured decomposition naturally applies, raising questions about whether this method generalizes to open-ended tasks.

[1] SuperCorrect: Advancing Small LLM Reasoning with Thought Template Distillation and Self-Correction

**Questions:**

1. The paper claims substantial improvements across models but fails to report essential inference parameters such as temperature, top-p sampling, this omission raises concerns about evaluation fairness and reproducibility. Different sampling configurations could significantly impact performance comparisons between methods, and without these details, it's unclear whether the reported gains reflect genuine methodological improvements or favorable hyperparameter choices.

2. The authors states existing self-correction methods "fail to extend to small models," but provide no clear definition of what constitutes a "small" model in this context. This ambiguity makes it difficult to assess the practical significance of the claimed contributions for genuinely small models.

3. The paper lacks analysis of inference cost, despite SELF-THOUGHT requiring multiple generation steps (initial answer, abstraction, refinement) compared to simpler baselines.

---

### Official Review · Reviewer_civL · 2025-11-01

**Soundness:** 3
**Presentation:** 3
**Contribution:** 2
**Rating:** 4
**Confidence:** 3

**Summary:**

This paper proposes SELF-THOUGHT, a prompt-based self-correction framework for large language models. Given an initial response, the model first distills the problem into a structured “task template” that captures essential variables, constraints, and objectives. This abstraction then guides the next-stage solution instantiation, producing refined answers grounded in problem understanding.
The authors also propose DISTIL-THOUGHT, which transfers task abstractions generated by larger models to smaller ones, enabling effective self-correction without costly fine-tuning.
Experiments on several reasoning benchmarks (e.g., AIME 2024/2025, Game of 24) show significant accuracy gains across both large and small models, outperforming existing self-correction approaches such as SELF-REFINE, REFLEXION, and SELF-TICK.

**Strengths:**

-  The idea of introducing a “task abstraction” stage before refinement is elegant and intuitively.

- The paper evaluates on diverse reasoning tasks and across multiple model sizes, showing consistent improvements over baselines.

- The proposed method achieve significant performance improvements without additional training.

- The DISTIL-THOUGHT variant provides a scalable and practical way to extend self-correction capabilities to smaller LLMs, which is a valuable contribution for cost-efficient deployment.

**Weaknesses:**

- This paper lacks formal analysis and case studies on why task abstraction leads to better correction beyond empirical evidence.

- Although the experiments show significant performance improvements, the proposed method essentially introduces only an additional prompt, with limited technical novelty and contribution.


- I notice that many baselines perform quite poorly in the experiments, and several even show substantial drops compared to the original model. This raises some concerns about the choice of baselines.

**Questions:**

- How sensitive is SELF-THOUGHT to the design of the abstraction prompt ℘?
- I’m curious about why this method achieves such remarkable performance gains, especially on Game of 24 and CheckmateInOne. Could the authors provide further explanations or illustrative case studies to clarify this?
- Whether the same abstraction template can generalize across different task families?

---

### Official Review · Reviewer_Kb8K · 2025-11-01

**Soundness:** 2
**Presentation:** 1
**Contribution:** 2
**Rating:** 2
**Confidence:** 4

**Summary:**

This paper proposes Self-Thought, a framework that introduces an intermediate step of task abstraction before solution refinement. It is claimed that Self-Thought improves accuracy, robustness, and generalization for both large and small models.

**Strengths:**

1. For significance, the paper idea proves to be effective on small LLMs.

**Weaknesses:**

1. I think the novelty of this paper is low, and the idea is very incremental. Your core idea is about prompting LLMs twice and using high-level summarization from the first run to help final reasoning. Performing summarization (and/or information extraction) as an additional step for reasoning enhancement has been extensively studied in applications such as RAG [1]. To me, this paper brings some incremental changes, but I do not get any interesting findings from reading it.

2. Soundness needs to be improved, especially on the baseline part. It is necessary to demonstrate that your "task distillation" is the optimal design compared to other summarization and information extraction baselines. For example, can you change the prompt to extract metadata first and then to perform reasoning? You may come up with several representative baselines accordingly to demonstrate whether your current implementation is optimal.

3. Improvement shrinks on reasoning-specific models such as R1, and this comes with the cost of running LLMs twice. Therefore, the contribution is limited, and I am not sure if this method is needed for large reasoning models. As recent LLMs are mostly reasoning capable, it is necessary to show "perform gain vs computation costs vs inference latency" on recent large reasoning models with and without your method, along with several other competitive baselines.

4. Clarity and writing need to be improved. For example, Figure 2 is not explicitly referenced/mentioned in the main content, so I have a hard time connecting it with your analysis. I think the pseudo-code in page 3 needs at least a caption. Moreover, some of the equations in Section 3 are probably unnecessary, lowering the overall readability.


[1] "RAPTOR: Recursive Abstractive Processing for Tree-Organized Retrieval"

**Questions:**

What is "task distillation"? Is it a new term proposed in this paper or are you referring to existing works? What is its difference compared to summarization and/or information extraction?

---

### Note · Authors · 2026-01-19

I have read and agree with the venue's withdrawal policy on behalf of myself and my co-authors.